# Synergistic enhancement of urban haze by nitrate uptake into transported hygroscopic particles in the Asian continental outflow

Jihoon Seo[1,2], Yong Bin Lim[3], Daeok Youn[4], Jin Young Kim[1], Hyoun Cher Jin[1]

[1]Environment, Health and Welfare Research Center, Korea Institute of Science and Technology, Seoul, 02792, South Korea
[2]School of Earth and Environmental Sciences, Seoul National University, Seoul, 08826, South Korea
[3]Departmant of Chemical Engineering and Material Science, Ewha Womans University, Seoul, 03760, South Korea
[4]Departmant of Earth Science Education, Chungbuk National University, Cheongju, 28644, South Korea

*Correspondence to*: Jin Young Kim (jykim@kist.re.kr)

**Abstract.** Haze pollution is affected by local air pollutants, regional transport of background particles and precursors, atmospheric chemistry related to secondary aerosol formation, and meteorological conditions conducive to the physical, dynamical, and chemical processes. In the large, populated and industrialized areas like the Asian continental outflow region, the combination of regional transport and local stagnation often exacerbates urban haze pollution. However, the detailed chemical processes underlying the enhancement of urban haze induced by the combined effect of local emissions and transported remote pollutants are still unclear. Here, we demonstrate an important role of transported hygroscopic particles in increasing local inorganic aerosols, by studying the chemical composition of PM$_{2.5}$ collected between October 2012 and June 2014 in Seoul, a South Korean megacity in the Asian continental outflow region, using the ISORROPIA II thermodynamic model. PM$_{2.5}$ measured under the condition of regional transport from the upwind source areas in China was higher in mass concentration and richer in secondary inorganic aerosol (SIA) species (SO$_4^{2-}$, NO$_3^-$, and NH$_4^+$) and aerosol liquid water (ALW) compared to that measured under the non-transport condition. The secondary inorganic species and ALW were both increased, particularly in cases with high PM$_{2.5}$ levels, and this indicates inorganic species as a major driver of hygroscopicity. We conclude that the urban haze pollution in the continental outflow region like Seoul, particularly during the cold season, can be exacerbated by ALW in the transported particles, which enhances the nitrate partitioning into the particle phase in NO$_x$- and NH$_3$-rich urban areas. This study reveals the synergistic effect of remote and local sources on urban haze pollution in the downwind region and provides insight into the nonlinearity of domestic and foreign contributions to receptor PM$_{2.5}$ concentrations in numerical air quality models.

## 1 Introduction

Fine particulate matter (under 2.5 μm in diameter; PM$_{2.5}$) in urban areas consists of inorganic species (SO$_4^{2-}$, NO$_3^-$, and NH$_4^+$) and organic matter (OM) produced by the gas-to-particle conversion of anthropogenic and biogenic precursors (Seinfeld and Pandis, 2016). These particles pose a public health problem due to their adverse effects on the human respiratory and cardiovascular systems (Pope and Dockery, 2006). Consequently, many countries have tried to mitigate urban haze pollution

by reducing local precursor emissions (van der A et al., 2017; Kim and Lee, 2018). However, since air pollution is also influenced by transported air pollutants, it is difficult to achieve improved air quality in megacities located in large, populated, and industrialized areas by controlling local emissions (Seo et al., 2018). Synoptic weather conditions cause stagnation and long-range transport that can lead to the accumulation of particles and gaseous precursors from local and remote sources. It can also change local meteorological factors to favorable conditions for secondary aerosol production, such as photo-oxidation and aqueous-phase processing (Sun et al., 2014; Zheng et al., 2015; Seo et al., 2017). Therefore, a better understanding of the combined effects of local emissions, regional transport, and meteorological conditions on urban haze pollution is required to establish appropriate mitigation strategies.

One key aerosol component that affects the formation and growth of haze particles is aerosol liquid water (ALW), which is ubiquitous and abundant in tropospheric fine particles (Nguyen et al., 2016). ALW not only increases the mass of secondary organic aerosols (SOA) by facilitating the partitioning of gas-phase water-soluble OM into the condensed phase, followed by aqueous-phase reactions (Asa-Awuku et al., 2010; Carlton and Turpin, 2013; McNeill, 2015; Marais et al., 2016), but also enhances the secondary inorganic aerosol (SIA) mass via nitrate formation through $HNO_3$ uptake and $N_2O_5$ hydrolysis (Zhang et al., 2015; Wang et al., 2017), and via sulfate production through the aqueous oxidation of $SO_2$ (Cheng et al., 2016; Wang et al., 2017). Studies of urban haze in the North China Plain reported simultaneous elevation of the relative humidity (RH), ALW, and SIA, which indicates the hygroscopic properties of inorganic species and the role of ALW in mass transfer into the particles (Liu et al., 2017; Tie et al., 2017; Wu et al., 2018). Combined with the ambient temperature and particle pH, ALW is critical for gas–particle partitioning of inorganic and water-soluble organic acid gases (Guo et al., 2018; Nah et al., 2018). Therefore, regional transport of wet particles to the precursor-rich urban environment will affect haze pollution downwind.

The South Korean capital city of Seoul and its metropolitan area is one of the highly populated megacities in East Asia, with a population of 25 million people, 9 million vehicles, and nearly half of the national gross domestic product, and has suffered from episodic haze events, particularly during the cold season (Seo et al., 2017; 2018). Although the Seoul metropolitan area is a large anthropogenic emission source of nitrogen oxides ($NO_x$), ammonia ($NH_3$), and volatile organic compounds (VOCs) (NIER, 2018), the effect of transported air pollutants from China cannot be ignored because of its location downwind from the major emission source region in China like the North China Plain and Yangtze River Delta (Fig. S1). In particular, severe multiday haze events in the Seoul metropolitan area mostly occur with a specific synoptic pattern, such as an eastward-moving high-pressure system, which induces regional transport of air pollutants from China and, subsequently, local stagnation (Seo et al., 2017; 2018). Therefore, the haze pollution in Seoul becomes severe primarily by accumulation of local and transported air pollutants. However, a recent numerical modeling study on regional contribution to the particulate concentration in Seoul reported a discrepancy between responses to the reduction of domestic and foreign emissions (Kim et al., 2017), and this implies additional production and growth of local haze particles by nonlinear interactions between local emissions and transported pollutants.

In this study, we explore the combined effects of local and remote sources, ALW and particle pH, and meteorological factors on the formation and growth of urban haze particles, based on daily measurement of $PM_{2.5}$ chemical compositions in Seoul,

backward trajectory analysis, and the ISORROPIA II thermodynamic model (Fountoukis and Nenes, 2007). Different chemical compositions and characteristics of Seoul haze according to the regional transport from China and the local stagnation in the Seoul metropolitan area is investigated from the perspective of inorganic partitioning and water uptake processes. An effective strategy for PM$_{2.5}$ reduction in Seoul is further discussed using ambient NO$_x$ and NH$_3$ levels, and analytic calculation of the HNO$_3$–NO$_3^-$ partitioning ratio as a function of ALW, pH, and temperature.

**2 Data and methods**

**2.1 Measurements and chemical analysis**

Daily PM$_{2.5}$ sampling was conducted on 210 days between October 2012 and June 2014 at the Korea Institute of Science and Technology (KIST) site in northeastern Seoul (37.603°N, 127.047°E, 58 m above sea level; Fig. S1). PM$_{2.5}$ samples for determining inorganic and carbonaceous species were collected on 47-mm Teflon filters (Pall Corporation, Port Washington,

NY, USA) with a Teflon-coated aluminum cyclone (URG Corporation, Chapel Hill, NC, USA) at a flow rate of 16.7 L/min, and on 203 × 254-mm quartz fiber filters (Whatman, Maidstone, UK) with a high-volume air sampler (flow rate of 1,000 L/min; Andersen Instruments, Atlanta, GA, USA), respectively.

The concentrations of inorganic ions (SO$_4^{2-}$, NO$_3^-$, Cl$^-$, NH$_4^+$, K$^+$, Ca$^{2+}$, Na$^+$, and Mg$^{2+}$) were measured using a 2000i/SP ion chromatograph (Dionex, Sunnyvale, CA, USA) after sonicating the Teflon filter sample for 30 min in a mixture of 0.5 mL of

ethanol and 14.5 mL of distilled deionized water. Using a piece of the quartz fiber filter sample (10 × 15 mm), the concentrations of organic carbon (OC) and elemental carbon (EC) were measured using a thermal/optical carbon aerosol analyzer (Sunset Laboratory, Tigard, OR, USA) based on National Institute for Occupational Safety and Health (NIOSH) method 5040 (Birch and Cary, 1996). We also identified 17 $n$-alkanes (C$_{20}$–C$_{36}$), 15 polycyclic aromatic hydrocarbons (PAHs; C$_{14}$–C$_{24}$), 19 monocarboxylic acids (C$_6$–C$_{20}$), 19 dicarboxylic acids (C$_3$–C$_{11}$), and 10 sugars (C$_5$–C$_6$, and C$_{12}$) using the extract

from one-half of the quartz fiber filter sample and a 7890A gas chromatograph (Hewlett Packard, Palo Alto, CA, USA) coupled to a 5975C mass selective detector (Agilent, Santa Clara, CA, USA). The OM / OC ratios derived from the measured OM and OC concentrations were used to estimate the total OM concentration. The organic compounds identified in this study constitute ~5% of the total OM. The analytical procedures used herein are described in detail elsewhere (Seo et al., 2017; Kim et al., 2018). In this study, out of 210 sampling days, 118 daily data that include not only inorganic species but also OM based on

the identified organic compounds' information were selected and utilized (Fig. S2).

Note that the PM$_{2.5}$ sampling on Teflon filter for inorganic ions was conducted without both a denuder and backup filters, and thus there could be potential sampling artifacts on the results, particularly negative artifacts in semivolatile ammonium nitrate (Ashbaugh and Eldred, 2004; Chow et al., 2005). Nie et al. (2010) reported that summertime nitrate loss on Teflon filter from the un-denuded filter sampling without backup filters is to be ~75% at lower nitrate concentrations (< 10 µg m$^{-3}$) but only

~10% at higher nitrate concentrations (> 10 µg m$^{-3}$) due to the formation of particle cake. Considering small evaporative loss in the cold season and the high nitrate concentration in Seoul, we expected small to moderate sampling errors in this study.

Sensitivity tests considering potential ammonium nitrate loss from the filter samples show that the assumption of 20% nitrate loss for the high concentrations with low-temperature groups and 50% nitrate loss for the low concentrations with moderate temperature group does not change our conclusion (Fig. S8).

The hourly concentrations of $SO_2$, $NO_2$, CO, $O_3$, and $PM_{10}$ at 34 air quality monitoring sites in Seoul (Fig. S1) provided by the Korea Ministry of Environment (KMOE) were averaged over all sites for each day, to obtain representative daily concentrations of each species for Seoul (Korea Environment Corporation, 2019). The hourly meteorological data of temperature, RH, wind speed, and solar irradiance at the Seoul weather station (37.571°N, 126.966°E) managed by the Korea Meteorological Administration (KMA) were averaged for each day and used in our analysis (KMA, 2019). Boundary layer

height (BLH) were derived from the European Centre for Medium-Range Weather Forecasts Reanalysis Interim (ERA-Interim) data (Dee et al., 2011; http://apps.ecmwf.int/datasets/data/interim-full-daily/, last access: 23 October 2019) at a grid point in Seoul (37.5°N, 127.0°E).

**2.2 Categorization of measurements**

To categorize the daily measurements according to (1) local atmospheric conditions (stagnation or ventilation) and (2) regional

impact (transport of pollutants from upwind source area), we used the 72-h backward trajectories from 500 m above the sampling site, obtained with the Hybrid Single-Particle Lagrangian Integrated Trajectory (HYSPLIT) model (https://ready.arl.noaa.gov; Stein et al., 2015) for every hour of each measurement day. We defined a local source area (Seoul metropolitan area) and two major upwind source areas in China (North China Plain and Yangtze River Delta) based on the satellite tropospheric $NO_2$ column density distribution (Fig. 1a), and calculated average residence time of daily 24 trajectories

in the Seoul metropolitan area ($t_{SMA}$) and the two source areas in China ($t_{CHN}$). The daily average residence time in each area shows different source characteristics such as the smaller sulfur-to-nitrogen emission ratio in Seoul (~0.06; 4.5 kt of $SO_x$ and 71.1 kt of $NO_x$ in 2010; NIER, 2018) compared with that of the Jing-Jin-Ji region in the North China Plain (~0.71; 2010 kt of $SO_2$ and 2830 kt of $NO_x$ in 2010; Li et al., 2017). For example, $t_{CHN}$ is highly correlated not only with particulate $SO_4^{2-}$ and $NO_3^-$ concentrations but also with their precursors ($SO_2$ and $NO_2$) concentrations, while $t_{SMA}$ has statistically significant

correlation only with $NO_2$ concentration (Fig. S3).

Since the medians of both $t_{SMA}$ and $t_{CHN}$ (for $t_{CHN} \neq 0$ h days) were ~6 h, we applied the 6 h as a reference trajectory residence time to categorize daily measurement data into relatively stronger and weaker influences by the local stagnation in the Seoul metropolitan area (with $t_{SMA}$) and the regional transport from China (with $t_{CHN}$). The measurement days were classified into one of four groups and summarized in Table 1: (i) the *local ventilation with no regional transport* (V-nT), in which the

trajectories did not originate from the source areas in China and also were not stagnant in the Seoul metropolitan area ($t_{SMA} <$ 6 h and $t_{CHN} = 0$ h; $n = 9$ days), (ii) the *local stagnation with no regional transport* (S-nT), in which the trajectories did not originate from the source areas in China but were stagnant in the Seoul metropolitan area ($t_{SMA} \geq 6$ h and $t_{CHN} = 0$ h; $n = 15$ days), (iii) the *local ventilation with regional transport* from the Chinese source area (V-T), in which the trajectories originated from the source areas in China but were not stagnant in the Seoul metropolitan area ($t_{SMA} < 6$ h and $t_{CHN} \geq 6$ h; $n = 15$ days),

and (iv) the *local stagnation with regional transport* from the Chinese source area (S-T), in which the trajectories originated from the source areas in China and also were stagnant in the Seoul metropolitan area ($t_{SMA} \geq 6$ h and $t_{CHN} \geq 6$ h; $n = 21$ days). The trajectory density distribution in the North China Plain and Yangtze River Delta areas were characterized for each group, with few trajectories for the *no regional transport* (V-nT and S-nT) groups but relatively dense trajectories for the *regional transport* (V-T and S-T) groups (Fig. 1b–d).

Despite the various potential factors like emissions, atmospheric chemistry, and meteorology that can affect the gas and particulate air quality, the simple categorization according to the residence time of backward trajectories could reveal different characteristics in chemical species and meteorological factors associated with the local stagnation and the regional transport of air pollutants. For example, the average concentrations of primary gaseous pollutants ($SO_2$, $NO_2$, and CO) of the *local stagnation with regional transport* (S-T) group are significantly higher than those of the *local stagnation with no regional*

*transport* (S-nT) group ($p \leq 0.001$), although average local meteorological factors between the two groups do not show significant differences (Table 2 and Figs. 2 and 3). On the other hand, the lower wind speed and shallower BLH were seen in the *local stagnation* (S-T and S-nT) groups compared with the *local ventilation* (V-T and V-nT) groups (Fig. 2), and the *local stagnation with regional transport* (S-T) group shows significantly higher levels of $SO_2$, $NO_2$, and CO in comparison with the *local ventilation with regional transport* (V-T) group, probably related to accumulation in the stagnant condition. Note that

although the present study used a part of total daily data that OM concentrations are available ($n = 118$), composite averages and differences of meteorological variables and air pollutant concentrations among the four categorized groups using the total daily data including the OM-unavailable dates ($n = 210$) showed the same characteristics as shown in Figs. 2 and 3 (Fig. S4).

**2.3 Prediction of ALW content and pH**

ISORROPIA II thermodynamic model was run in forward mode to estimate ALW content associated with inorganic species,

pH, and the equilibrium gas–particle partitioning based on the daily $PM_{2.5}$ ionic compositions, RH, and temperature. ISORROPIA II uses total (gas- plus aerosol-phase) measurements as inputs, under the metastable assumption (no solid precipitates). Recent validation studies demonstrate that forward mode is influenced less by measurement errors and gives a more accurate pH than reverse mode, which uses only the aerosol-phase composition as an input (Hennigan et al., 2015; Song et al., 2018).

The forward mode of ISORROPIA II requires the total $NH_3$ (gas-phase $NH_3$ plus particulate $NH_4^+$), total $HNO_3$ ($HNO_3$ plus particulate $NO_3^-$), and total Cl (gas-phase HCl plus particulate $Cl^-$) concentrations, as well as the particulate $SO_4^{2-}$, $K^+$, $Ca^{2+}$, $Na^+$, and $Mg^{2+}$ concentrations. However, the ambient $NH_3$, $HNO_3$, and HCl data were not available for this study. To overcome this problem with the input data, we used statistically reconstructed $NH_3$ data and then estimated $HNO_3$ using a thermodynamic model.

Firstly, the daily $NH_3$ concentrations in Seoul from January 2012 to December 2014 were reconstructed using the statistical characteristics of a year-long record of $NH_3$ at the Gwangjin site (37.545°N, 127.096°E) in Seoul for the preceding year (346 days between September 2010 and August 2011; Phan et al., 2013). There were small interannual changes in the estimated

NH$_3$ emission in Seoul (Fig. S5; NIER, 2018), and both two sites share similar environments located downwind from the downtown core under the prevailing westerlies and surrounded by residential and small urban green areas (Fig. S1). Therefore, we assumed that the statistical characteristics of both two sites and periods would be similar despite the temporal gap (~1–3 yr) and spatial distance (~7.8 km) between this study and Phan et al. (2013). We built a multiple linear regression model of NH$_3$ that retains statistical characteristics of the measured NH$_3$ such as the annual average and standard deviation of 10.9 ± 4.25 ppb and the significant linear correlations ($p < 0.05$) with temperature, RH, wind speed, and SO$_2$, NO$_2$, and CO concentrations (Text S1 and Table S1). The reconstructed NH$_3$ concentration is seasonally high during the warm season as reported by Phan et al. (2013) and shows high peaks on the polluted days with stagnant condition in the cold season (Fig. S6). In the next step, the ISORROPIA II calculation was performed, using the reconstructed NH$_3$ and measured NH$_4^+$ as the total equivalent NH$_3$, and the measured NO$_3^-$ alone as the total HNO$_3$, to estimate the HNO$_3$–NO$_3^-$ partitioning ratio. Then, using both the measured NO$_3^-$ and the ratio between the HNO$_3$ and NO$_3^-$ predicted from the initial result, we estimated the ambient HNO$_3$. Note that we regarded the HNO$_3$ / NO$_3^-$ ratio as approximately independent of the total HNO$_3$. Although the nitrate partitioning depends on temperature, ALW content, and pH in general, the HNO$_3$ / NO$_3^-$ ratio is affected alone by pH because both temperature and ALW are independent variables in this step. In the existence of excess NH$_3$ as we assumed for Seoul (~10 ppb at the first step), changes in pH by total HNO$_3$ are limited owing to the buffering effect of NH$_3$–NH$_4^+$ partitioning (Weber et al., 2016; Guo et al., 2018; Lim et al., 2020). The pH, nitrate partitioning ratio [$\varepsilon(NO_3^-) = NO_3^- / (HNO_3 + NO_3^-)$], and ammonium partitioning ratio [$\varepsilon(NH_4^+) = NH_4^+ / (NH_3 + NH_4^+)$] from the ISORROPIA simulations with and without gas-phase HNO$_3$ information show negligible differences between each other (Figs. 4d–f), unlike the significant role of additional NH$_3$ information on increasing pH and $\varepsilon(NO_3^-)$ and decreasing $\varepsilon(NH_4^+)$ (Figs. 4a–c).

Finally, we repeated the ISORROPIA simulation with both total NH$_3$ and total HNO$_3$ estimated from the previous steps. Here we did not consider HCl–Cl$^-$ partitioning because the Cl$^-$ fraction in dry PM$_{2.5}$ (~1% in average) is significantly smaller than the NO$_3^-$ and NH$_4^+$ fractions (~18% and ~12% in average, respectively).

Although there are uncertainties in the reconstructed NH$_3$ and HNO$_3$ due to lack of direct measurements, their impact on the estimation of inorganic ALW and particle pH is small enough. The good agreement between the predicted and statistically reconstructed NH$_3$ concentrations ($R^2 \sim 0.95$), as well as between the predicted and measured SO$_4^{2-}$, NO$_3^-$, and NH$_4^+$ concentrations ($R^2 > 0.95$; Fig. S7), ensures small errors on water contents of individual inorganic species. Since water uptake by inorganic aerosol in the ISORROPIA II is based on the Zdanovskii-Stokes-Robinson (ZSR) mixing rule (Stokes and Robinson, 1966), which approximates the water content of a mixture as a sum of water contents of individual salts at the same RH (Fountoukis and Nenes, 2007), expected error in $W_i$ induced by potential errors in NH$_3$ and HNO$_3$ would also be small. Regarding pH, buffering effect of semivolatile NH$_3$ partitioning reduces sensitivity of pH to excess NH$_3$ in the NH$_3$-rich conditions like Seoul (Weber et al., 2016; Guo et al., 2018; Lim et al., 2020). We conducted ISORROPIA simulations with various NH$_3$ levels (0–50 ppb) applied for all the measurement days as fixed values to further explore the sensitivity of pH, $\varepsilon(NO_3^-)$, and $\varepsilon(NH_4^+)$ to NH$_3$. Good correlations between results from the daily varied NH$_3$-simulation and the fixed NH$_3$-simulations shows that potential errors induced by the estimation of daily NH$_3$ levels will be small if NH$_3$ concentrations are

nonzero (Figs. 5a, c, and e). Increasing pH and $\varepsilon(NO_3^-)$ and decreasing $\varepsilon(NH_4^+)$ by the increase in fixed $NH_3$ level become flatten over the 5 ppb of $NH_3$ (Figs. 5b, d, and f), and this indicates that the potential errors in reconstructed $NH_3$ will not significantly change the results in this study.

Since ALW content associated with OM ($W_o$) is not considered in the ISORROPIA II, we estimated it according to the $\kappa$-Köhler theory and the ZSR mixing rule (Nguyen et al., 2015, and references therein) by the following equation:

$$W_o = V_o \kappa_{org} \frac{a_w}{1-a_w} \tag{1}$$

where $V_o$ is volume of OM ($\mu m^3\ cm^{-3}$), which is obtained from OM concentration ($\mu g\ m^{-3}$) divided by a typical organic density of 1.4 g cm$^{-3}$ (Turpin and Lim, 2001), $\kappa_{org}$ is hygroscopicity parameter (dimensionless) calculated from the parameterized

relationship of $\kappa_{org} = (0.29 \pm 0.05) \cdot (O/C)$ for the range of O / C from 0.3 to 0.6 (Chang et al., 2010), and $a_w$ is water activity (dimensionless) that is assumed to be equivalent to RH (Nguyen et al., 2016).

## 3 Results and discussion

### 3.1 Effects of regional transport and local stagnation on PM$_{2.5}$ composition

Average concentrations of PM$_{2.5}$ and chemical components were highest in the *local stagnation with regional transport* (S-T)

group (e.g., PM$_{2.5\ dry}$ of 72.2 $\mu g\ m^{-3}$) and lowest in the *local ventilation with no regional transport* (V-nT) group (e.g., PM$_{2.5\ dry}$ of 19.9 $\mu g\ m^{-3}$), and the average concentrations for the *local ventilation with regional transport* (V-T) group and the *local stagnation with no regional transport* (S-nT) groups (e.g., PM$_{2.5\ dry}$ of 53.3 $\mu g\ m^{-3}$ and 34.4 $\mu g\ m^{-3}$, respectively) located between that of the S-T and V-nT groups (Table 2 and Fig. 3f). The composite difference between the *regional transport* (V-T and S-T) groups and *no regional transport* (V-nT and S-nT) groups reveals the regional effect of transported haze particles

and precursors from the North China Plain and Yangtze River Delta, while that between the *local stagnation* (S-T and S-nT) groups and *local ventilation* (V-T and V-nT) groups shows the effect of accumulation or diffusion of both local and transported pollutants in the Seoul metropolitan area. Thus, the significant difference of each PM$_{2.5}$ component between the S-T and V-nT groups (Fig. 6) indicates both contributions of local stagnant condition over the Seoul metropolitan area and regional transport from the Chinese source area to the exacerbation of haze pollution in Seoul.

The average concentrations of gaseous precursors like $SO_2$ and $NO_2$ for the S-T group were about 2 times higher than those for the V-nT group (Fig. 3), while $SO_4^{2-}$, $NO_3^-$, and $NH_4^+$, and inorganic ALW increased by 5–10 times (Fig. 6). In contrast, average OM concentration for the S-T group was 2 times higher than that for the V-nT group similarly to CO, and organic ALW of the S-T group showed a relatively smaller increase (~4 times of the V-nT group) than inorganic ALW (~8 times of the V-nT group). Therefore, the PM$_{2.5}$ increase in Seoul seems to be induced not only by the physical and dynamical processes

like transport or accumulation of air pollutants but also by the chemical process like secondary aerosol formation, particularly related to the increase in inorganic species and ALW.

Statistically significant difference ($p < 0.05$) between the S-T and S-nT groups (effect of regional transport in the local stagnant condition) was observed in SIA species ($SO_4^{2-}$, $NO_3^-$, and $NH_4^+$), OM, and ALW (both $W_i$ and $W_o$), and that between the S-T and V-T groups (effect of local stagnation in the regional transport condition) was seen for EC and OM (Table 2). Thus, although both regional transport and local stagnation can induce high concentrations of PM$_{2.5}$ and its chemical components of the S-T group, the effect of regional transport is more significant for the increase in inorganic species and water content, while the effect of local stagnation is more significant for the increase in carbonaceous species. Sulfur oxidation ratio [SOR = $SO_4^{2-}$ / ($SO_2 + SO_4^{2-}$)], nitrate partitioning ratio [$\varepsilon(NO_3^-)$], and ammonium partitioning ratio [$\varepsilon(NH_4^+)$] of the S-T group, which were significantly higher than that of the S-nT group ($p < 0.05$) but did not clearly differ from that of the V-T group, indicate that the increase in SIA by the regional transport effect is closely associated with the enhanced oxidation (e.g., from $NO_x$ to $HNO_3$) or partitioning of inorganic species into the particle phase.

Note that OM concentration seems to be increased by both effects of regional transport and local stagnation (Table 2 and Fig. 6f). However, the average estimates of O / C ratio (~0.55) and OM / OC ratio (~1.87), which are in between semi-volatile oxygenated OA (SV-OOA) and low-volatility oxygenated OA (LV-OOA), are similar for the four categorized groups (Table 2). Together with weak correlations of O / C ratio with wind speed ($r = -0.093$) and BLH ($r = -0.172$) in total daily data ($n = 118$), this suggests that aging and oxidation of SOA in Seoul was not much dependent on external conditions like stagnation or transport. Rather, significant correlations of O / C ratio with temperature ($r = -0.531$) and solar irradiance ($r = -0.370$) indicate the winter-high / summer-low seasonality of the secondary fraction in OA. Such temperature-dependent SOA fraction can be also found in OC / EC ratio ($r = -0.638$ for temperature and $r = -0.469$ for solar irradiance), which can be regarded as an index for secondary OC, and is consistent with partitioning theory (Takekawa et al., 2003).

## 3.2 Fractional changes of ALW and SIA by effects of regional transport and local stagnation

The chemical mass fraction of PM$_{2.5}$ are expected to be affected by source characteristics (emissions) and chemical process (secondary aerosol formation), not likely by the physical and dynamical processes (transport and accumulation), which affect the mass concentration of PM$_{2.5}$. Thus, the mass fraction of each component is helpful to understand the source and chemical characteristics of each categorized group.

In terms of the component fraction, PM$_{2.5}$ of the *local stagnation with regional transport* (S-T) group can be characterized by inorganic-rich, wet particles, while that of the *local ventilation with no regional transport* (V-nT) group was relatively organic-richer and drier (Fig. 7). Estimated wet PM$_{2.5}$ (including ALW of both $W_i$ and $W_o$; PM$_{2.5 \text{ wet}}$) of the S-T group contained larger fractions of SIA species ($SO_4^{2-}$, $NO_3^-$, and $NH_4^+$) and ALW but a smaller OM fraction (46% SIA, 23% OM, and 26% ALW) than that of the V-nT group (29% SIA, 43% OM, and 15% ALW). Comparisons with the *local ventilation with regional transport* (V-T) group (37% SIA, 19% OM, and 33% ALW) and the *local stagnation with no regional transport* (S-nT) group (34% SIA, 35% OM, and 21% ALW) groups show that the effect of regional transport increases inorganic and water fractions, while the effect of local stagnation raises organic fraction of the total PM$_{2.5 \text{ wet}}$. Note that the increase in ALW fraction from the V-nT (15%) to V-T (33%) groups is much larger than that from the S-nT (21%) to S-T (26%) groups because of the

significantly higher increase in RH from the V-nT (46%) to V-T (61%) groups compared to that from the S-nT (54%) to S-T (52%) groups. The relationship between RH and ALW, as well as hygroscopic properties of the Seoul haze particles, will be further discussed in the next section.

The $SO_4^{2-}$ fraction of dry $PM_{2.5}$ (excluding $W_i$ and $W_o$; $PM_{2.5\ dry}$) was ~17% for the *no regional transport* (V-nT and S-nT) groups and about 20% and 23% for the V-T and S-T groups, respectively. The $SO_4^{2-}$ fraction in the *no regional transport* groups is close to the background fraction of $SO_4^{2-}$ in $PM_{2.5\ dry}$ in Seoul, and this fraction is comparable to the "local sulfate source" contribution of 21.7% (mostly in the form of ammonium sulfate and equivalent to the $SO_4^{2-}$ contribution of ~16%) identified by the previous source apportionment study on $PM_{2.5}$ in Seoul (Kim et al., 2016). A previous field study conducted on the multiday haze episode showed that the $SO_4^{2-}$ concentration at the upwind background site over the Yellow Sea (Deokjeok Island; 37.233°N, 126.149°E) was nearly the same as the Seoul $SO_4^{2-}$ at the regional-transport stage of haze or the clean period, and even kept ~50% of the Seoul $SO_4^{2-}$ concentration at the local-stagnation stage of the haze episode (Seo et al., 2017). Thus, together with considering the small $SO_2$ emission in the Seoul metropolitan area compared to the North China Plain (Li et al., 2017; NIER, 2018) and the $SO_2$ increase by the regional transport effect (*regional transport* groups minus *no regional transport* groups; ~2–4 ppb), the majority of $SO_4^{2-}$ in the *regional transport* (V-T and S-T) groups seems to be directly transported from China or produced during the transport from China before arriving at the Seoul metropolitan area.

The $NO_3^-$ fraction of $PM_{2.5\ dry}$ was only ~9% for the V-nT group, but increased to 16% and 23% for the S-nT and V-T groups, and reached up to more than 24% for the S-T group. Interestingly, the fractional increase by the local stagnation effect (*local stagnation* groups minus *local ventilation* groups; ~2–7%) is much smaller than that by the regional transport effect (*regional transport* groups minus *no regional transport* groups; ~9–14%) despite the same increase in $NO_2$ concentration (~15 ppb; Table 2) by two different effects. Seoul is one of the largest sources of $NO_x$ in Korea, but the previous study using satellite $NO_x$ column data with the Lagrangian model demonstrated that the Chinese contribution to the $NO_2$ columns over Korea reaches up to ~50–70% (Lee et al., 2014). However, the impact of regional transport on the increase in $NO_3^-$ fraction, which is ~2 times higher than that of the local stagnation, indicates a nonlinear effect of regional transport on the $NO_3^-$ increase in the Seoul haze.

The $NH_4^+$ fraction of $PM_{2.5\ dry}$ was 8% for the V-nT group, 10% for the S-nT group, 13% for the V-T group, and 15% for the S-T group. The higher fractional increase of $NH_4^+$ by the regional transport effect (T- minus nT-groups; 5%) compared to that by the local stagnation effect (S- minus V-groups; 2%) results from the higher increase in both $SO_4^{2-}$ and $NO_3^-$ aerosols related to the regional transport effect.

**3.3 Synergistic effect of transported hygroscopic particle on SIA in cold season**

The high-concentration, inorganic-rich, and wet particles (high SIA and ALW fractions of $PM_{2.5\ wet}$) mostly belong to the *regional transport* (V-T and S-T) groups, of which backward trajectories originated from the upwind regional source area (the North China Plain and Yangtze River Delta), while relatively low-concentration, organic-rich dry particles (high OM fraction of $PM_{2.5\ wet}$) belong to the *local ventilation with no regional transport* (V-nT) group (Figs. 8a and b). The ALW fraction is a

function of RH and also increases with the SIA fraction at the same RH (Fig. 8c), similar to the Beijing haze (Wu et al., 2018). This indicates that the Seoul haze is hygroscopic and the major driver of its hygroscopicity is inorganic species. Since the SIA fraction was relatively higher in the *regional transport* groups compared to that in the *no regional transport* groups, the hygroscopic properties of PM$_{2.5}$ in Seoul were more clearly revealed in the *regional transport* groups (Fig. 8d). Therefore, ALW fraction in the transported particles (e.g., the *regional transport* groups) was higher than that in the particles of local origin (e.g., the *local stagnation with no regional transport* (S-nT) group).

The concentrations of SO$_4^{2-}$, NO$_3^-$, and NH$_4^+$ increase with their gas-phase precursors (SO$_2$, NO$_2$, and NH$_3$) in Seoul, except the V-nT group. However, the regression slopes of the SO$_4^{2-}$, NO$_3^-$, and NH$_4^+$ with respect to their precursors in the *regional transport* groups are steeper than the slope of the S-nT group (Fig. 9). Such higher sensitivity of increase in SO$_4^{2-}$, NO$_3^-$, and NH$_4^+$ to the gas-phase precursors of the *regional transport* groups is related to the higher ratios of oxidation (SOR) and partitioning [$\varepsilon$(NO$_3^-$) and $\varepsilon$(NH$_4^+$)] compared to the S-nT group (Figs. 10b, f, and j). Since the average concentrations of total sulfur (= SO$_2$ + SO$_4^{2-}$), total HNO$_3$, and total NH$_3$ are highest in the *local stagnation with regional transport* (S-T) group due to combination of remote precursors transported from China and local precursors accumulated in the stagnant condition (Figs. 10a, e, and i), the high SOR, $\varepsilon$(NO$_3^-$), and $\varepsilon$(NH$_4^+$) of the *regional transport* groups induced the highest average SO$_4^{2-}$, NO$_3^-$, and NH$_4^+$ concentrations of the S-T group (Table 2).

The SOR, $\varepsilon$(NO$_3^-$), and $\varepsilon$(NH$_4^+$) averaged for four range classes of temperature and inorganic ALW ($W_i$) show that all the ratios increased with $W_i$ (Figs. 10d, h, and l), and $\varepsilon$(NO$_3^-$) and $\varepsilon$(NH$_4^+$) decreased with temperature (Figs. 10g and k). Changes in $W_i$ from the S-nT group (~7 µg m$^{-3}$ in average; $W_{i\,1}$) to the *regional transport* groups (> 20 µg m$^{-3}$ in average; $W_{i\,3}$) increase SOR, $\varepsilon$(NO$_3^-$), and $\varepsilon$(NH$_4^+$) by approximately up to 20% for each ratio. This implies that the transported haze particles in the *regional transport* groups, which are richer in inorganics and more hygroscopic than the local haze particles of the S-nT group (Fig. 8), promote SIA production. The aqueous-phase oxidation of SO$_2$ via H$_2$O$_2$, O$_3$, NO$_2$, and transition metal ion (TMI) pathways (Harris et al., 2013; Cheng et al., 2016; Wang et al., 2016), and the partitioning of HNO$_3$ and NH$_3$ via dissolution and dissociation in the wet particles (Seinfeld and Pandis, 2016), are effective ways to produce SO$_4^{2-}$, NO$_3^-$, and NH$_4^+$. Therefore, ALW in the hygroscopic particles can synergistically increase the inorganic species in the precursor-rich urban area like Seoul.

In terms of the synergistic increase in NO$_3^-$ with ALW, the ratio between NO$_3^-$ and SO$_4^{2-}$ can be an important factor. Hygroscopic uptake of ALW by both SO$_4^{2-}$ and NO$_3^-$ can increase pH by dilution effect on hydrogen ions (H$^+$). Because NO$_3^-$ is a semi-volatile hygroscopic species, the higher pH increased by ALW allows more partitioning of HNO$_3$ gas into the particle phase and uptakes more ALW. However, if SO$_4^{2-}$ is dominant in the particle, such a feedback process will be weakened because SO$_4^{2-}$ is non-volatile (Guo et al., 2017). The average NO$_3^-$-to-SO$_4^{2-}$ molar ratios of the *regional transport* groups (1.87 for the S-T group and 1.60 for the V-T group) are higher than that of the *no regional transport* groups (1.29 for the S-nT group and 0.81 for the V-nT group). Since ALW, pH, and $\varepsilon$(NO$_3^-$) in the *regional transport* (V-T and S-T) groups are higher than those in the *no regional transport* (V-nT and S-nT) groups, this feedback process can explain the synergistic effect of transported particle on the high NO$_3^-$ and ALW fractions.

Note that the partitioning of HNO$_3$ and NH$_3$ is also dependent on temperature ($T$) (Figs. 10g and k). Changes in $T$ from the S-nT group (~15°C in average; $T_3$) to the *regional transport* groups (~9°C in average; $T_2$) also can increase $\varepsilon(\text{NO}_3^-)$ and $\varepsilon(\text{NH}_4^+)$ by approximately 20% for each, similar to the effect of inorganic ALW. In Korea, the regional transport of air pollutants from the NCP area is usually accompanied by westerly or northwesterly continental outflow, which induces cold advection, during the cold season (Kim et al., 2018). The colder temperature of the S-T group (~9°C) compared to the S-nT group (~15°C) could help the higher sensitivity of the SIA increase to the precursor concentrations in the S-T group (Fig. 9). Interestingly, the increasing SOR by temperature (and also irradiance) is not significant as much as that by inorganic ALW (Figs. 10c) despite the high-temperature and strong-irradiance conditions conducive to photochemical oxidation of SO$_2$ in summer. This implies that the observed high SO$_4^{2-}$ in the S-T group resulted not only from the photochemical gas-phase oxidation but also considerably from the aqueous-phase oxidation of SO$_2$ in the transported wet particles.

### 3.4 Effect of NO$_x$ and NH$_3$ control on mitigating haze pollution in Seoul

The $\varepsilon(\text{NO}_3^-)$ can be analytically calculated according to the solubility and dissociation of gas-phase HNO$_3$, and represented as a sigmoid function (S-curve) of pH (Guo et al., 2018; Nah et al., 2018) by the following equation:

$$\varepsilon(\text{NO}_3^-) = \frac{H^*_{\text{HNO}_3} W_i RT (0.987 \times 10^{-14})}{\gamma_{\text{H}^+} \gamma_{\text{NO}_3^-} 10^{-\text{pH}} + H^*_{\text{HNO}_3} W_i RT (0.987 \times 10^{-14})} \tag{2}$$

where $0.987 \times 10^{-14}$ is the unit conversion factor from atm and μg to SI, $R$ is the gas constant (8.314 J mol$^{-1}$ K$^{-1}$), $W_i$ (μg m$^{-3}$) is the ALW content associated with inorganic species, $H^*_{\text{HNO}_3}$ is the effective Henry's law constant (products of Henry's law constant for HNO$_3$ gases and the acid dissociation constant for HNO$_3 \leftrightarrow \text{NO}_3^- + \text{H}^+$; mol$^2$ kg$^{-2}$ atm$^{-1}$) dependent on temperature and pH (Clegg et al., 1998). The product of activity coefficients, $\gamma_{\text{H}^+} \gamma_{\text{NO}_3^-}$, is dependent on both temperature and ionic strength (Kusik and Meissner, 1978; Kim et al., 1993). In this study, daily $\gamma_{\text{H}^+} \gamma_{\text{NO}_3^-}$ were obtained based on daily average ambient temperature and ionic strength from ISORROPIA II, and ranged from 0.135 to 0.734, with an average of 0.333 ± 0.150.

The S-curve of $\varepsilon(\text{NO}_3^-)$ as a function of pH is useful for examining the role of inorganic ALW and temperature in HNO$_3$–NO$_3^-$ partitioning and provides insights into the potential HNO$_3$ partitioning by change in particle pH. In Fig. 11, both decrease in temperature and increase in ALW can shift the $\varepsilon(\text{NO}_3^-)$ curves toward a lower pH, so more HNO$_3$ can exist in the particle phase. Almost 100% of the total HNO$_3$ exists in the particle phase at the higher pH (> 4) in Seoul. Since the gas-phase HNO$_3$ largely formed by the reaction of NO$_2$ with hydroxyl radicals (OH) is highly water-soluble (Seinfeld and Pandis, 2016), the transported wet particles (the *regional transport* (V-T and S-T) groups) can easily take up HNO$_3$ gas into the aerosol solution in the NO$_x$-rich (~60 ppb) environment of Seoul (Vellingiri et al., 2015), particularly during the cold season. Note that ALW can additionally enhance the nitrate radical (NO$_3$)–N$_2$O$_5$ pathway and heterogeneous N$_2$O$_5$ hydrolysis (Bertram et al., 2009). In addition, the high NH$_3$ level (~10 ppb) in Seoul (Phan et al., 2013) helps further to increase the uptake of HNO$_3$ gas into the aerosol solution (Guo et al., 2018).

To reduce ammonium nitrate aerosols in Seoul, therefore, two ways can be suggested; the $NO_x$ control and the $NH_3$ control. As shown in Fig. 9b, the observed $NO_3^-$ concentration is highly correlated with the $NO_2$ level, and this implies that the $NO_x$ control will be an effective way to reduce total $HNO_3$ and $NO_3^-$ concentrations. Such a direct relationship between $NO_x$ control and nitrate aerosol is significant at the condition of $\varepsilon(NO_3^-)$ close to 1. Because of the higher $\varepsilon(NO_3^-)$ of the transported wet particles, the $NO_x$ reduction will more effectively reduce $NO_3^-$ in the *regional transport* groups compared to the *local stagnation with no regional transport* (S-nT) group. For example, the potential $NO_3^-$ reduction by the 10 ppb decrease of ambient $NO_2$ concentration deduced from the linear regression in Fig. 9b can be estimated as ~8 µg m$^{-3}$ for the *local ventilation with regional transport* (V-T) group, ~7 µg m$^{-3}$ for the *local stagnation with regional transport* (S-T) group, and ~5 µg m$^{-3}$ for the *local stagnation with no regional transport* (S-nT) group, respectively. On the other hand, the $NH_3$ control to achieve the low $\varepsilon(NO_3^-)$ by lowering the particle pH from the current level (pH of ~3.5 in average) for the S-T group may cost more than that for the S-nT group, because the $\varepsilon(NO_3^-)$ in the S-nT group (green circles) starts to decrease rapidly at pH < 3.5, while that in the *regional transport* groups (yellow circles for the V-T group and red circles for the S-T group) remains relatively high (~80%) at the lower pH (~2.5) (Fig. 11). Since the haze pollution in Seoul usually becomes severe with the regional transport of hygroscopic wet particles (e.g., the S-T group), more reduction of the $NH_3$ emissions should be required for the higher-concentration S-T group compared to the lower-concentration S-nT group. However, the benefit of reducing PM$_{2.5}$ mass concentration by $NH_3$ control can be cancelled out by the adverse effects of strong particle acidity on human health (Fang et al., 2017).

## 4 Conclusions

Based on PM$_{2.5}$ chemical speciation, gaseous pollutants, and meteorological data in Seoul together with backward trajectory analysis, the present study investigated chemical compositions and characteristics of urban haze particle in the Asian continental outflow region, according to physical and dynamical conditions such as local stagnant condition in the urban area and regional transport of air pollutants from the remote source area. Although various factors like local emissions of primary pollutants and secondary precursors, atmospheric chemistry related to the secondary formation and aging of aerosols, and meteorological conditions can affect the urban haze pollution, a simple categorization by average residence times of backward trajectories within the local (Seoul metropolitan area) and remote (North China Plain and Yangtze River Delta) source areas showed clearly distinguishable characteristics in concentration and fractional composition of PM$_{2.5}$ among the local stagnation / ventilation groups and the regional transport / non-transport groups. In particular, ALW content associated with inorganic species and particle pH estimated by the ISORROPIA II thermodynamic model helped to show different hygroscopic and inorganic partitioning properties of Seoul haze by the categorized groups.

The measurement group of *local stagnation with regional transport* (S-T) from China is characterized by higher PM$_{2.5}$ concentration (72 ± 32 µg m$^{-3}$ of PM$_{2.5\ dry}$) and more inorganic-rich and wetter particles (46% SIA, 23% OM, and 26% ALW in PM$_{2.5\ wet}$) in comparison to the measurement group of *local ventilation with no regional transport* (V-nT; 20 ± 5 µg m$^{-3}$ of

$PM_{2.5\ dry}$ and 29% SIA, 43% OM, and 15% ALW in $PM_{2.5\ wet}$). Increase in SIA from the *local ventilation with no regional transport* (V-nT) group to the *local stagnation with regional transport* (S-T) group (~7 times from ~7 to 45 µg m$^{-3}$) is larger than the OM increase (~2 times from 10 to 23 µg m$^{-3}$) but relates closely to the increase in inorganic ALW (~8 times from ~3 to 22 µg m$^{-3}$), indicating inorganic species as a major driver of hygroscopicity. The larger increase in SIA species ($SO_4^{2-}$ from ~3 to 17 µg m$^{-3}$ and $NO_3^-$ from ~2 to 18 µg m$^{-3}$) compared to the increase in gaseous precursors ($SO_2$ from 5 to 9 ppb and

$NO_2$ from 27 to 57 ppb) suggests that there is not only the accumulation of local and transported particles but also additional chemical processes for the SIA production in the combination of regional transport and local stagnant conditions.

    Further comparisons with the *local stagnation with no regional transport* (S-nT) group and the *local ventilation with regional transport* (V-T) group shows the stronger influence of regional transport rather than local stagnation on the high oxidation and partitioning ratios [SOR, $\varepsilon(NO_3^-)$, and $\varepsilon(NH_4^+)$] associated with the SIA increase. SOR, $\varepsilon(NO_3^-)$, and $\varepsilon(NH_4^+)$ were raised up

to ~20% by the increase in inorganic ALW in the Seoul haze condition, and this demonstrates an important role of ALW of the transported hygroscopic particles in increasing of SIA fraction. In addition, $\varepsilon(NO_3^-)$ and $\varepsilon(NH_4^+)$ are decreased by temperature, and thus the transported wet particles can efficiently convert the gas-phase $HNO_3$ and $NH_3$ into the particle-phase $NO_3^-$ and $NH_4^+$ during the cold season. Therefore, the synergistic effect of transported wet particles and local precursors on the SIA increase and high $PM_{2.5}$ concentration in Seoul can be most prominent in the cold season, when the continental outflow

dominates East Asia and helps transboundary transport of air pollutants to Korea. Since $SO_2$ emissions in Seoul are small, the increased $SO_4^{2-}$ in the regional transport condition is likely to be transported from the remote source areas in China or produced during the transport. On the other hand, the high-$NO_x$ (~60 ppb) and $NH_3$-rich (~10 ppb) conditions in Seoul can promote the uptake of $HNO_3$ into the wet particle.

    Most of the severe haze events in Seoul occur in the local stagnant conditions combined with transport of the preceding regional

haze (Seo et al., 2017). The transported regional haze particles with a high inorganic fraction and abundant ALW readily take up $HNO_3$ and $NH_3$ gases under the high $NO_x$ and $NH_3$ conditions of the urban area and consequently reduce the air quality more than local haze formation without regional transport (Fig. 12). Considering both the high $\varepsilon(NO_3^-)$ of the transported wet particles and the low pH required to decrease $\varepsilon(NO_3^-)$ in the combined regional transport–local stagnation condition, $NO_x$ control rather than $NH_3$ control may be a more effective $PM_{2.5}$ reduction strategy in Seoul.

Our results provide insight into the nonlinear effects of the transported particles and local precursors on urban haze pollution in the regional air quality modeling system. For example, domestic and foreign contributions to $PM_{10}$ concentration over the Seoul metropolitan area estimated by the brute force method approach shows a discrepancy between reductions of domestic and foreign emissions, in particular during the cold season (Kim et al., 2017). The synergistic enhancement of urban haze pollution by the combination of regional and local sources makes precise estimation of domestic and foreign contributions

difficult. This study also shows need for international cooperation in air quality management.

## Code availability

ISORROPIA II is available at: https://isorropia.epfl.ch/code-repository/. HYSPLIT model is available at: https://www.ready.noaa.gov/HYSPLIT.php.

## Data availability

Daily $PM_{2.5}$ measurements, chemical analysis, meteorological factors and ISORROPIA II results data utilized in this study are available at: https://drive.google.com/open?id=1hJrwViP_qx23dTuBxLapbWhAn5LrTXMI. The hourly data of $SO_2$, $NO_2$, CO, $O_3$, and $PM_{10}$ concentrations at 34 air quality monitoring sites in Seoul for the analysis period are available on the website managed by the Korea Environment Corporation (2020; https://www.airkorea.or.kr/web/last_amb_hour_data?pMENU_NO=123). The hourly meteorological data of temperature, sea

level pressure, relative humidity, wind speed, and solar irradiance at the Seoul weather station for the same period can be found on the website of the KMA (2020; https://data.kma.go.kr/data/grnd/selectAsosRltmList.do?pgmNo=36). The ERA-Interim data (Dee et al., 2011) can be accessed via the European Centre for Medium-Range Weather Forecasts (ECMWF) data server (http://apps.ecmwf.int/datasets/data/interim-full-daily/).

## Author contributions

JS initiated the investigation and performed thermodynamic modeling analyses. JS and YBL extensively discussed the concept. HCJ conducted the field measurement and provided chemical analyses. DY and JYK provided additional feedback on the manuscript. JS prepared the manuscript with contributions from all co-authors.

## Competing interests

The authors declare that they have no conflict of interest.

**Acknowledgements**

This research was supported by the Korea Institute of Science and Technology (KIST) and the National Strategic Project - Fine Particle of the National Research Foundation of Korea (NRF) funded by the Ministry of Science and ICT (MSIT), the Ministry of Environment (ME), and the Ministry of Health and Welfare (MOHW) (2017M3D8A1090654). Yong Bin Lim was supported by NRF (2019M3D8A1070941). Daeok Youn was supported by the Basic Science Research Program through the

National Research Foundation of Korea (NRF) funded by the Ministry of Education (2015R1D1A3A01020130) and Korea Environment Industry & Technology Institute (KEITI) funded by Korea Ministry of Environment (MOE) (2018001310004).

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

**Table 1: List of four categorized groups of daily PM$_{2.5}$ measurements in this study.**

| Categories | Acronyms | Trajectory residence time in the Chinese source area ($t_{CHN}$) | Trajectory residence time in the Seoul metropolitan area ($t_{SMA}$) |
|---|---|---|---|
| *Local ventilation with no regional transport* | V-nT | $t_{CHN} = 0$ h | $t_{SMA} < 6$ h |
| *Local stagnation with no regional transport* | S-nT | $t_{CHN} = 0$ h | $t_{SMA} \geq 6$ h |
| *Local ventilation with regional transport* | V-T | $t_{CHN} \geq 6$ h | $t_{SMA} < 6$ h |
| *Local stagnation with regional transport* | S-T | $t_{CHN} \geq 6$ h | $t_{SMA} \geq 6$ h |

**Table 2: The average and standard deviation of PM$_{2.5}$ chemical composition, meteorological factors, related gas concentrations, thermodynamic model results and gas-particle partitioning ratios, and mass concentration ratios to wet PM$_{2.5}$ (PM$_{2.5\,wet}$) for the *local ventilation with no regional transport* (V-nT) group, the *local stagnation with no regional transport* (S-nT) group, the *local ventilation with regional transport* (V-T) group, and the *local stagnation with regional transport* (S-T) group, and *p*-values derived from Welch's *t*-test for the composite differences between the S-T and S-nT groups and between the S-T and V-T groups.**

| Components | Units | Composite average and standard deviation | | | | *p*-values from Welch's *t*-test | |
| --- | --- | --- | --- | --- | --- | --- | --- |
| | | [V-nT] ($n = 9$) | [S-nT] ($n = 15$) | [V-T] ($n = 15$) | [S-T] ($n = 21$) | [S-T] minus [S-nT] (Transport effect) | [S-T] minus [V-T] (Stagnation effect) |
| Meteorological factors | | | | | | | |
| $T$ | (°C) | $2.6 \pm 13.2$ | $15.2 \pm 11.5$ | $9.4 \pm 6.0$ | $9.3 \pm 8.5$ | $p = 0.104$ | $p = 0.967$ |
| RH | (%) | $45.9 \pm 10.0$ | $53.5 \pm 10.2$ | $61.0 \pm 9.3$ | $51.8 \pm 11.7$ | $p = 0.632$ | $p = 0.013$ |
| WS | (m s$^{-1}$) | $3.5 \pm 0.7$ | $2.4 \pm 0.6$ | $3.2 \pm 0.9$ | $2.4 \pm 0.7$ | $p = 0.824$ | $p = 0.009$ |
| SI | (W m$^{-2}$) | $167 \pm 41$ | $172 \pm 53$ | $154 \pm 64$ | $137 \pm 67$ | $p = 0.094$ | $p = 0.445$ |
| BLH [a] | (m) | $770 \pm 177$ | $550 \pm 210$ | $666 \pm 277$ | $457 \pm 165$ | $p = 0.165$ | $p = 0.016$ |
| Gaseous species | | | | | | | |
| SO$_2$ | (ppb) | $4.6 \pm 0.6$ | $5.2 \pm 2.0$ | $6.6 \pm 2.2$ | $8.8 \pm 2.7$ | $p < 0.001$ | $p = 0.012$ |
| NO$_2$ | (ppb) | $27.4 \pm 4.1$ | $41.6 \pm 11.7$ | $41.2 \pm 9.5$ | $57.3 \pm 12.5$ | $p < 0.001$ | $p < 0.001$ |
| CO | (ppm) | $0.42 \pm 0.05$ | $0.55 \pm 0.26$ | $0.66 \pm 0.28$ | $0.89 \pm 0.31$ | $p = 0.001$ | $p = 0.026$ |
| O$_3$ | (ppb) | $24.6 \pm 7.0$ | $20.1 \pm 11.9$ | $23.1 \pm 10.5$ | $15.5 \pm 9.0$ | $p = 0.219$ | $p = 0.031$ |
| NH$_3$ [b] | (ppb) | $4.9 \pm 4.1$ | $10.5 \pm 3.0$ | $10.2 \pm 3.4$ | $11.0 \pm 4.3$ | $p = 0.668$ | $p = 0.499$ |
| PM$_{2.5}$ components | | | | | | | |
| PM$_{2.5\,dry}$ | (μg m$^{-3}$) | $19.9 \pm 4.8$ | $34.4 \pm 20.7$ | $53.3 \pm 33.7$ | $72.2 \pm 31.9$ | $p < 0.001$ | $p = 0.100$ |
| SO$_4^{2-}$ | (μg m$^{-3}$) | $3.3 \pm 0.7$ | $5.8 \pm 4.1$ | $10.8 \pm 7.6$ | $16.7 \pm 11.2$ | $p < 0.001$ | $p = 0.069$ |
| NO$_3^-$ | (μg m$^{-3}$) | $1.7 \pm 1.4$ | $5.4 \pm 7.3$ | $12.1 \pm 11.2$ | $17.6 \pm 12.1$ | $p < 0.001$ | $p = 0.166$ |
| NH$_4^+$ | (μg m$^{-3}$) | $1.6 \pm 0.5$ | $3.4 \pm 3.6$ | $6.9 \pm 5.7$ | $10.8 \pm 6.9$ | $p < 0.001$ | $p = 0.075$ |
| EC | (μg m$^{-3}$) | $1.4 \pm 0.6$ | $1.7 \pm 0.6$ | $1.6 \pm 0.6$ | $2.1 \pm 0.8$ | $p = 0.128$ | $p = 0.036$ |
| OM | (μg m$^{-3}$) | $10.1 \pm 4.4$ | $15.3 \pm 10.1$ | $15.0 \pm 7.0$ | $22.9 \pm 8.6$ | $p = 0.025$ | $p = 0.005$ |
| $W_i$ | (μg m$^{-3}$) | $2.7 \pm 1.4$ | $7.4 \pm 7.5$ | $23.7 \pm 24.9$ | $22.2 \pm 18.6$ | $p = 0.003$ | $p = 0.853$ |
| $W_o$ | (μg m$^{-3}$) | $0.9 \pm 0.4$ | $2.2 \pm 1.7$ | $3.1 \pm 2.7$ | $3.3 \pm 2.3$ | $p = 0.087$ | $p = 0.833$ |
| Carbonaceous analysis | | | | | | | |
| O / C | (atomic ratio) | $0.55 \pm 0.11$ | $0.55 \pm 0.07$ | $0.54 \pm 0.04$ | $0.55 \pm 0.07$ | $p = 0.883$ | $p = 0.622$ |
| OM / OC | (mass ratio) | $1.87 \pm 0.15$ | $1.87 \pm 0.10$ | $1.86 \pm 0.05$ | $1.87 \pm 0.08$ | $p = 0.927$ | $p = 0.632$ |
| OC / EC | (mass ratio) | $4.2 \pm 1.7$ | $4.8 \pm 2.4$ | $5.7 \pm 2.5$ | $6.2 \pm 2.1$ | $p = 0.071$ | $p = 0.510$ |
| ISORROPIA II analysis | | | | | | | |
| Ionic strength | (M) | $54.3 \pm 16.2$ | $42.2 \pm 17.6$ | $30.6 \pm 10.3$ | $47.1 \pm 22.7$ | $p = 0.470$ | $p = 0.006$ |
| pH | | $3.1 \pm 0.6$ | $3.2 \pm 0.7$ | $3.5 \pm 0.5$ | $3.5 \pm 0.7$ | $p = 0.186$ | $p = 0.808$ |
| Oxidation / Partitioning | | | | | | | |
| SOR | (molar ratio) | $0.13 \pm 0.04$ | $0.19 \pm 0.07$ | $0.25 \pm 0.09$ | $0.28 \pm 0.12$ | $p = 0.006$ | $p = 0.439$ |
| $\varepsilon(NO_3^-)$ | (molar ratio) | $0.83 \pm 0.28$ | $0.67 \pm 0.31$ | $0.93 \pm 0.09$ | $0.90 \pm 0.19$ | $p = 0.017$ | $p = 0.481$ |
| $\varepsilon(NH_4^+)$ | (molar ratio) | $0.40 \pm 0.27$ | $0.25 \pm 0.14$ | $0.43 \pm 0.18$ | $0.52 \pm 0.18$ | $p < 0.001$ | $p = 0.150$ |

[a] Reanalysis data from the ERA-Interim at 37.5°N, 127.0°E. [b] Statistically reconstructed data based on Phan et al. (2013).

$T$: temperature; RH: relative humidity; WS: wind speed; SI: solar irradiance; BLH: boundary layer height; EC: elemental carbon; OM: organic matter; $W_i$: aerosol liquid water (ALW) content associated with inorganic species; $W_o$: ALW content associated with OM; OC: organic carbon; SOR: sulfur oxidation ratio; $\varepsilon(NO_3^-)$: nitrate partitioning ratio; $\varepsilon(NH_4^+)$: ammonium partitioning ratio.

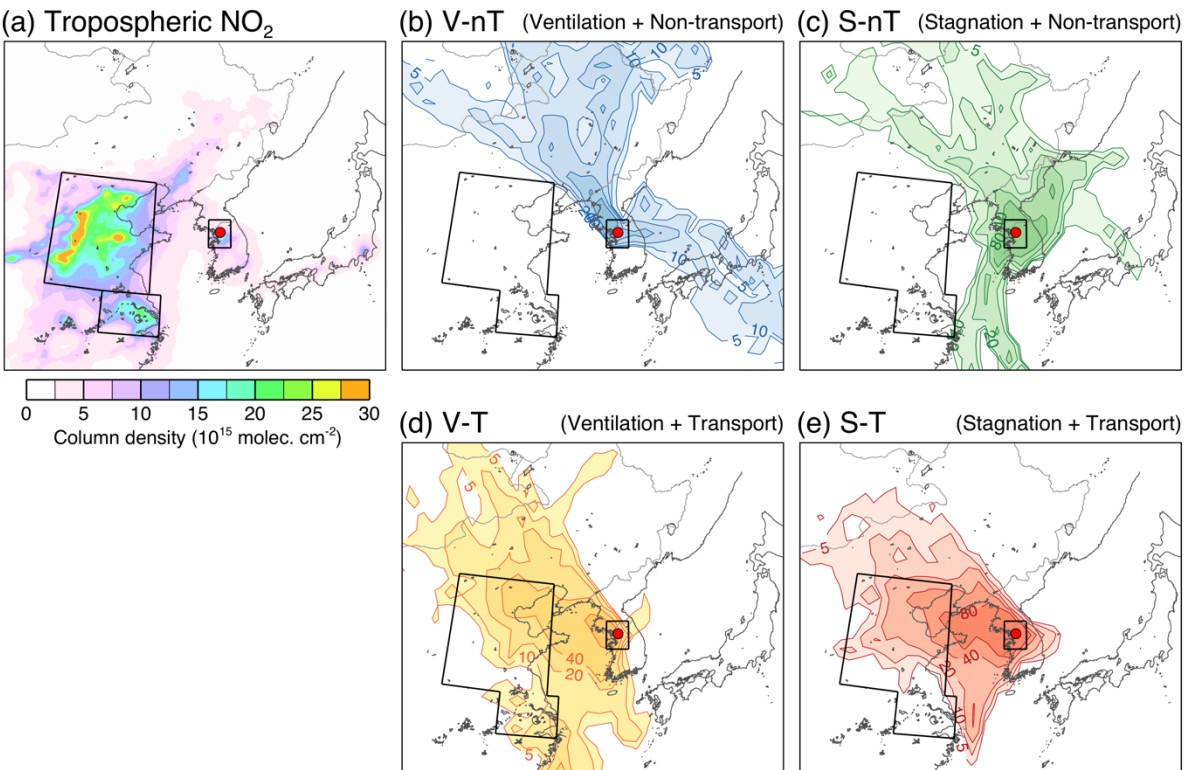

**Figure 1: (a)** Tropospheric NO$_2$ column density obtained from the Ozone Monitoring Instrument (OMI) onboard the EOS-Aura satellite, averaged for 2012–2014, and the major anthropogenic emission areas defined in this study (North China Plain [NCP]: 112°E –121°E, 33°N –41°N; Yangtze River Delta [YRD]: 117°E –122°E, 30°N –33°N; and Seoul metropolitan area [SMA]; 126°E – 128°E, 36.5°N –38.5°N). **(b–e)** Trajectory frequency (number of endpoints in each 1° × 1° grid cell per number of trajectories, %) derived from the HYSPLIT 72-h backward trajectories obtained 500 m above the Korea Institute of Science and Technology (KIST) site and average chemical compositions of particulate matter under 2.5 μm in diameter (PM$_{2.5}$) for the **(b)** *local ventilation with no regional transport* (V-nT) group, **(c)** *local stagnation with no regional transport* (S-nT) group, **(d)** *local ventilation with regional transport* (V-T) group, and **(e)** *local stagnation with regional transport* (S-T) group. Seoul is marked with a solid red circle.



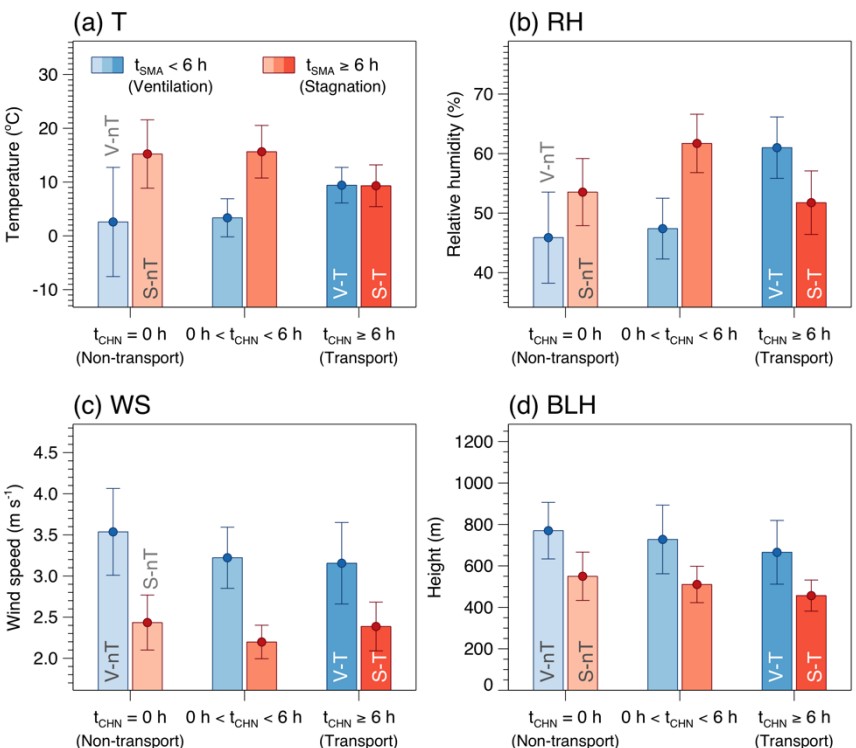

**Figure 2:** Averages and confidence intervals at 95% of (a) temperature (*T*), (b) relative humidity (RH), (c) wind speed (WS), and (d) boundary layer height (BLH) for the six case groups categorized by the ranges of the average daily residence time of backward trajectories in the Seoul metropolitan area ($t_{SMA}$) and in the North China Plain and Yangtze River Delta ($t_{CHN}$).

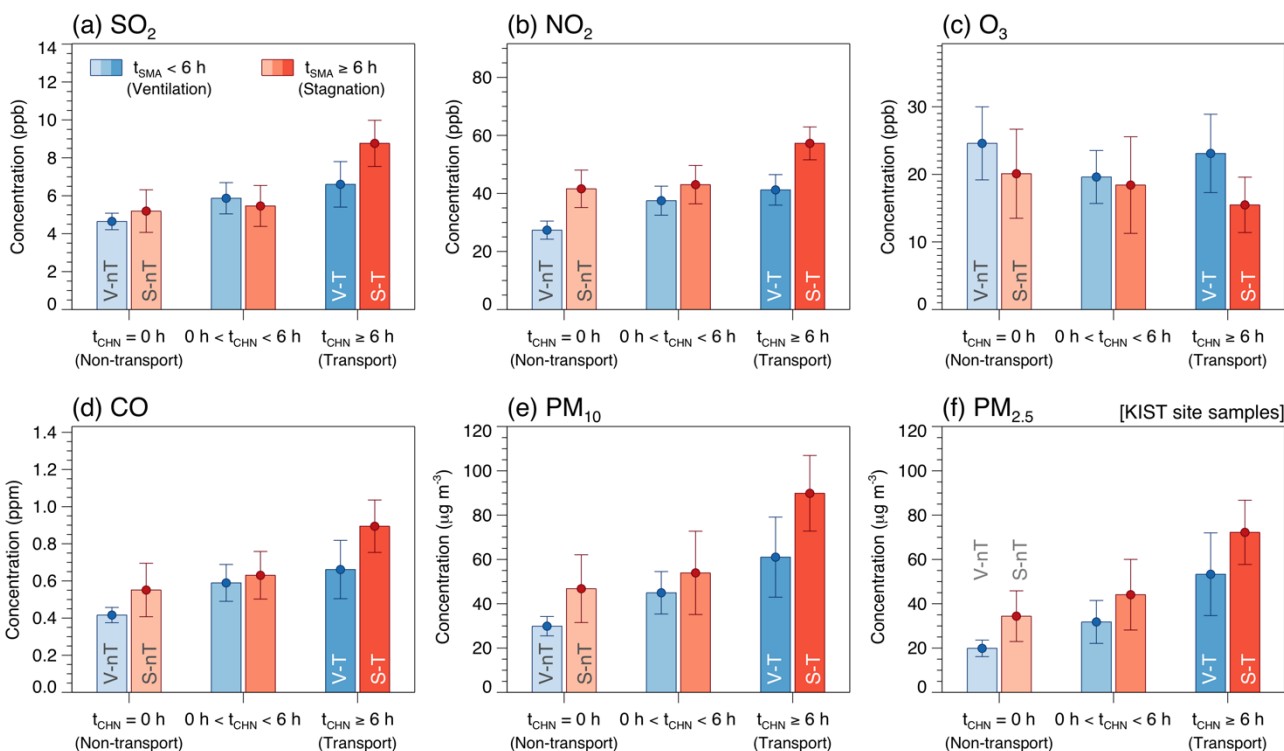


**Figure 3: Averages and confidence intervals at 95% of daily average concentrations of (a) SO₂, (b) NO₂, (c) O₃, (d) CO, and (e) PM₁₀, together with (f) PM₂.₅ measured at the KIST site in this study (dry PM₂.₅) for the six case groups categorized by ranges of the average daily residence time of backward trajectories in the Seoul metropolitan area ($t_{SMA}$) and in the North China Plain and Yangtze River Delta ($t_{CHN}$).**


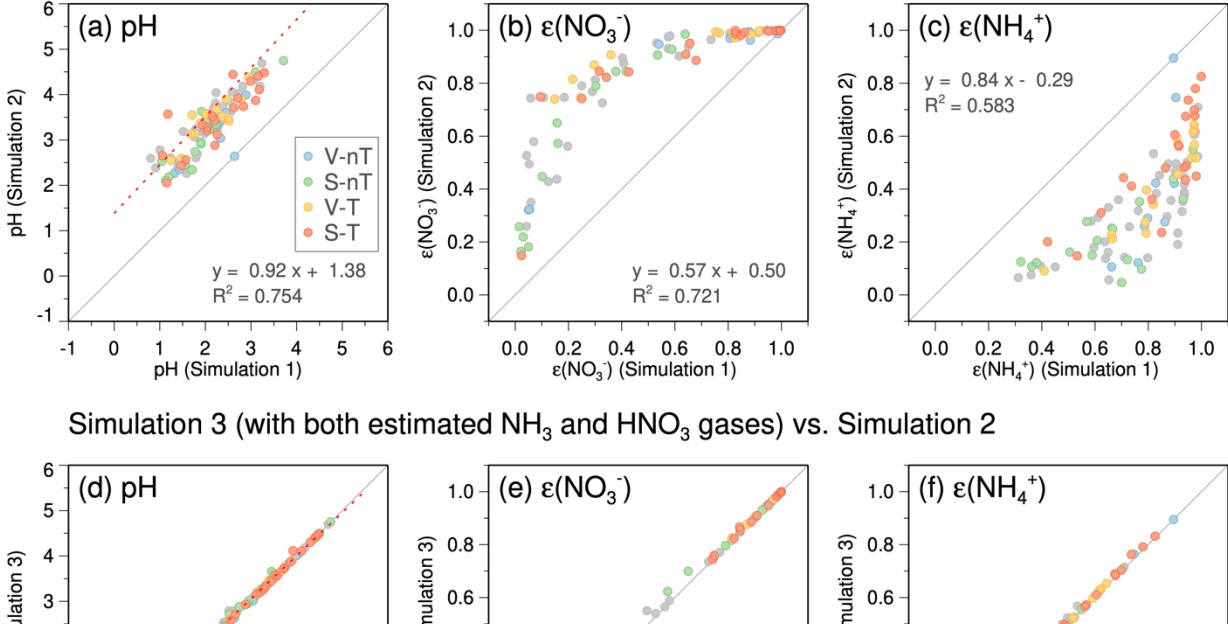

**Figure 4: Comparisons of the predicted pH, nitrate partitioning ratio [$\varepsilon(NO_3^-) = NO_3^- / (HNO_3 + NO_3^-)$], and ammonium partitioning ratio [$\varepsilon(NH_4^+) = NH_4^+ / (NH_3 + NH_4^+)$]** between (a–c) the simulation without gas-phase $NH_3$ and $HNO_3$ information (using $NH_4^+$ and $NO_3^-$ as the total $NH_3$ and total $HNO_3$; Simulation 1) and the simulation with reconstructed $NH_3$ (using $NH_3 + NH_4^+$ as the total $NH_3$ but only $NO_3^-$ as the total $HNO_3$; Simulation 2) and (d–f) the Simulation 2 and the simulation with both estimated gas-phase $NH_3$ and $HNO_3$ information (this study; Simulation 3). Filled circles in light blue, light green, light yellow, and light red colors represent daily data belong to the *local ventilation with no regional transport* (V-nT) group, *local stagnation with no regional transport* (S-nT) group, *local ventilation with regional transport* (V-T) group, and *local stagnation with regional transport* (S-T) group, respectively. Gray solid lines indicate a 1-to-1 relationship.

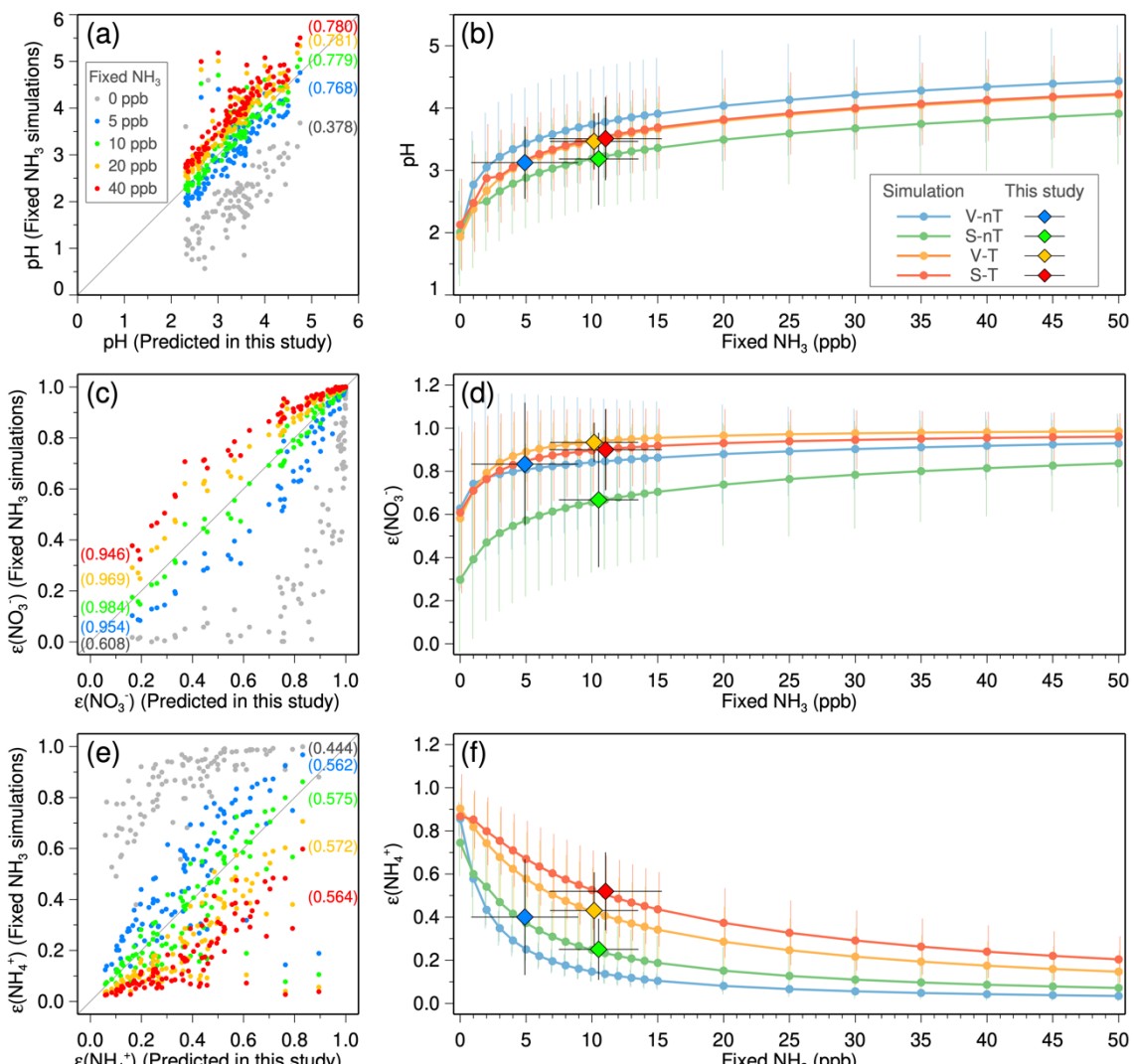

**Figure 5: Sensitivity of (a–b) predicted pH, (c–d) nitrate partitioning ratio [$\varepsilon(NO_3^-) = NO_3^- / (HNO_3 + NO_3^-)$], and (e–f) ammonium partitioning ratio [$\varepsilon(NH_4^+) = NH_4^+ / (NH_3 + NH_4^+)$] to gas-phase ammonia (NH$_3$) concentrations. (a, c, and e) Comparisons between the simulation in this study (using the daily reconstructed NH$_3$ concentrations) and the simulations with the various fixed NH$_3$ levels (colored dots in gray, blue, green, yellow, and red for 0 ppb, 5 ppb, 10 ppb, 20 ppb, and 40 ppb, respectively). Values in round brackets are the R-squared values of the linear regressions. Gray solid lines indicate a 1-to-1 relationship. (b, d, and f) Average (solid circles) and standard deviation (vertical bars) of pH, $\varepsilon(NO_3^-)$, and $\varepsilon(NH_4^+)$ for the *local ventilation with no regional transport* (V-nT) group (light blue), *local stagnation with no regional transport* (S-nT) group (light green), *local ventilation with regional transport* (V-T) group (light yellow), and *local stagnation with regional transport* (S-T) group (light red) with respect to the various fixed NH$_3$ levels from 0 ppb to 50 ppb. Averages and standard deviations of the reconstructed NH$_3$ concentrations and obtained pH, $\varepsilon(NO_3^-)$, and $\varepsilon(NH_4^+)$ for the V-nT, S-nT, V-T, and S-T groups are represented as blue, green, yellow, and red diamonds with horizontal and vertical bars.**

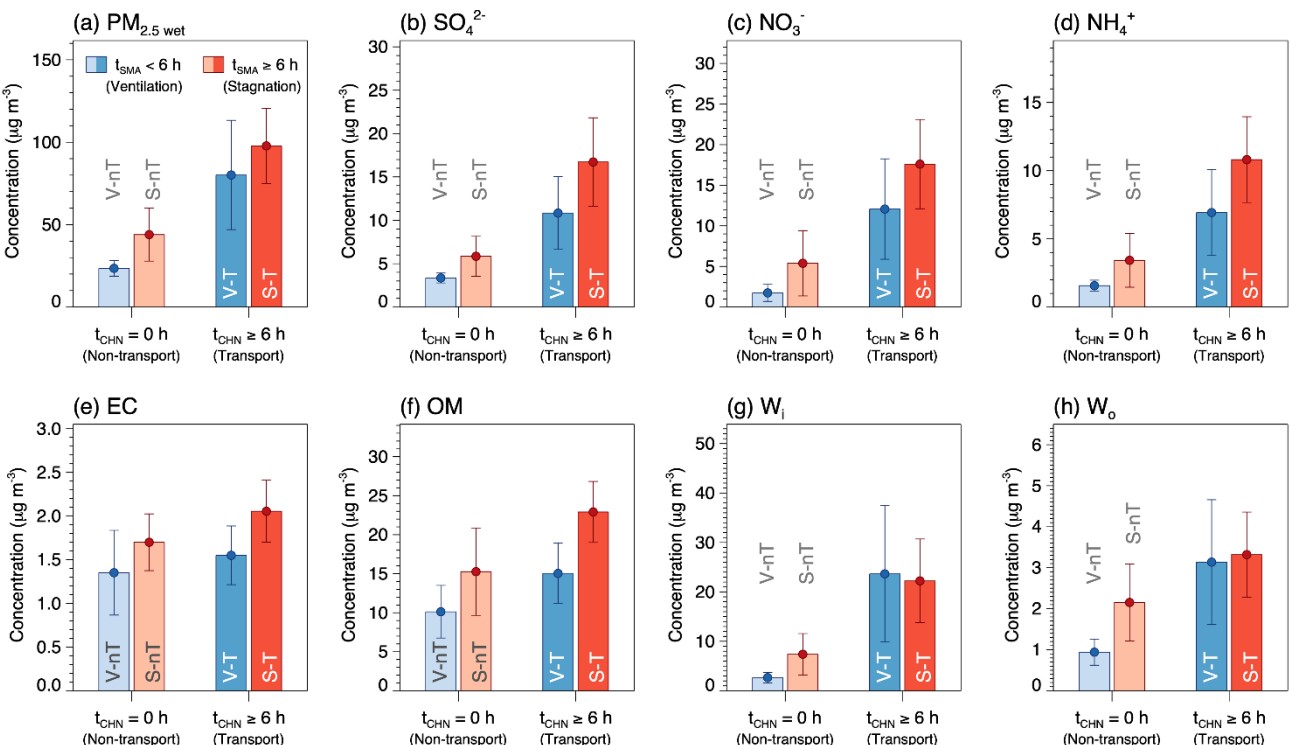

**Figure 6: Average concentrations and confidence intervals at 95% of (a) wet PM$_{2.5}$ (PM$_{2.5 \, wet}$), (b) sulfate (SO$_4^{2-}$), (c) nitrate (NO$_3^-$), (d) ammonium (NH$_4^+$), (e) elemental carbon (EC), (f) organic matter (OM), (g) inorganic ALW ($W_i$), and (h) organic ALW ($W_o$) for the** *local ventilation with no regional transport* **(V-nT) group (light blue),** *local stagnation with no regional transport* **(S-nT) group (light red),** *local ventilation with regional transport* **(V-T) group (darker light blue), and** *local stagnation with regional transport* **(S-T) group (darker light red) categorized by ranges of the average daily residence time of backward trajectories in the Seoul metropolitan area ($t_{SMA}$) and in the North China Plain and Yangtze River Delta ($t_{CHN}$).**

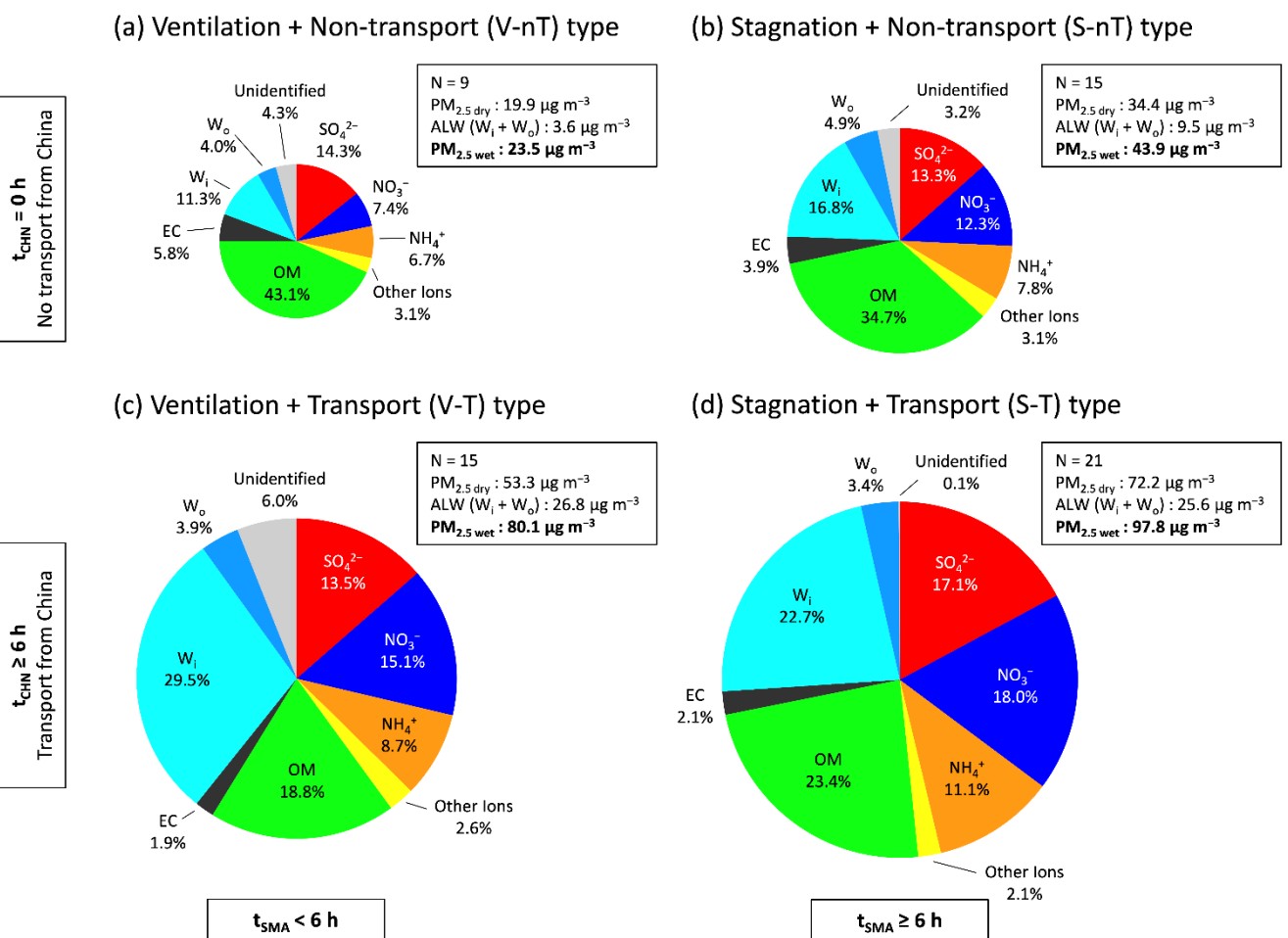

**Figure 7: Average chemical compositions of wet PM$_{2.5}$ (PM$_{2.5\,wet}$) for the (a) *local ventilation with no regional transport* (V-nT) group, (b) *local stagnation with no regional transport* (S-nT) group, (c) *local ventilation with regional transport* (V-T) group, and (d) *local stagnation with regional transport* (S-T) group. Fractional species are sulfate (red), nitrate (dark blue), ammonium (orange), organic matter (green), elemental carbon (black), inorganic ALW (light blue), organic ALW (darker light blue), and other ions (chloride, sodium, potassium, magnesium, and calcium; yellow). The size of each circle is proportional to the PM$_{2.5\,wet}$ concentration.**

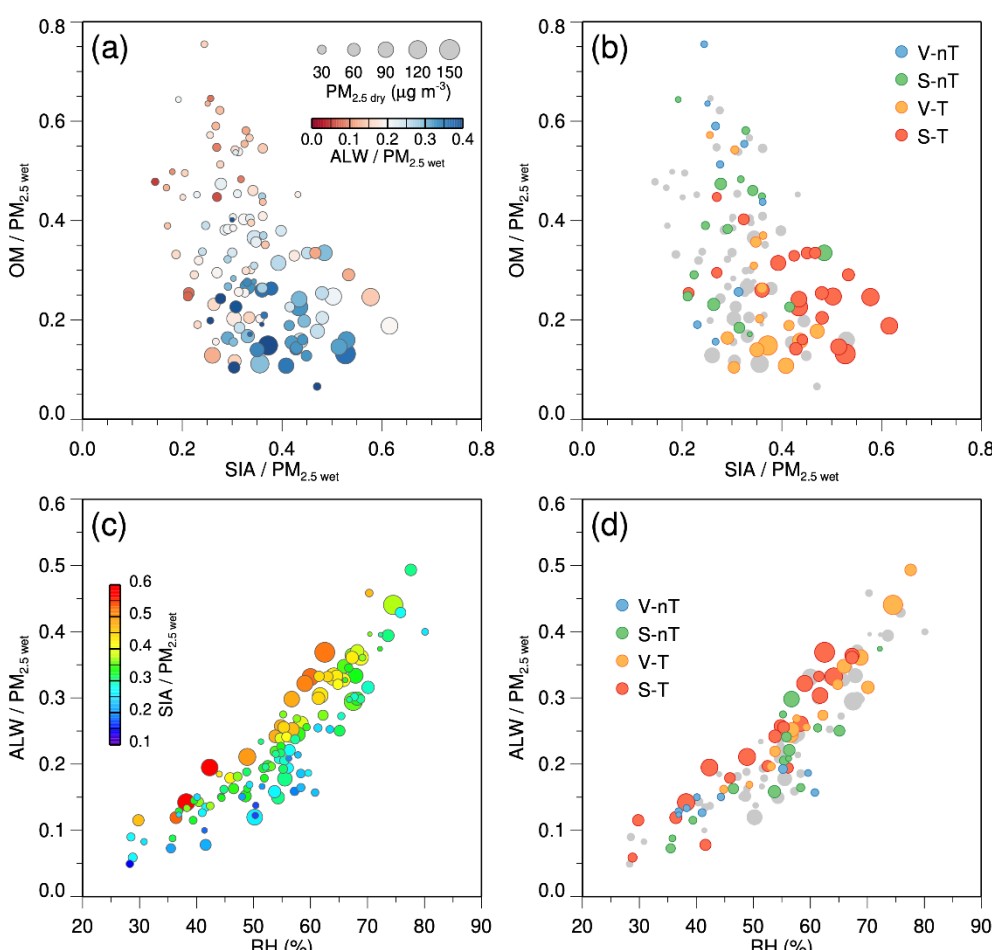

Figure 8: (a–b) Scatterplots of the secondary inorganic aerosol (SIA) fraction to wet PM$_{2.5}$ (SIA / PM$_{2.5 \text{ wet}}$) ratio versus the organic matter (OM) fraction to wet PM$_{2.5}$ (OM / PM$_{2.5 \text{ wet}}$) ratio, colored according to (a) the aerosol liquid water (ALW) to wet PM$_{2.5}$ (ALW / PM$_{2.5 \text{ wet}}$) ratio and (b) the categories for each case. (c–d) Scatterplots of ALW / PM$_{2.5 \text{ wet}}$ versus relative humidity (RH), colored according to (c) SIA / PM$_{2.5 \text{ wet}}$ ratio and (d) the categories. The size of each circle is proportional to the dry PM$_{2.5}$ (PM$_{2.5 \text{ dry}}$) concentration.

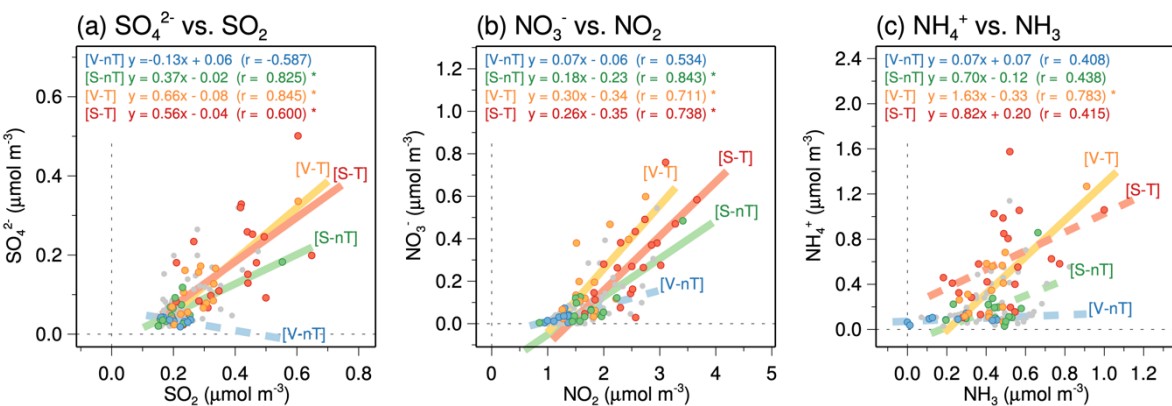

**Figure 9:** Scatterplots of (a) SO$_4^{2-}$ versus SO$_2$, (b) NO$_3^-$ versus NO$_2$, and (c) NH$_4^+$ versus the reconstructed ammonia (NH$_3$). Filled circles in blue, green, yellow, and red colors represent daily data belong to the *local ventilation with no regional transport* (V-nT) group, *local stagnation with no regional transport* (S-nT) group, *local ventilation with regional transport* (V-T) group, and *local stagnation with regional transport* (S-T) group, respectively. The statistically significant linear regression slopes (*p* < 0.01) for each group were represented by solid lines.

680

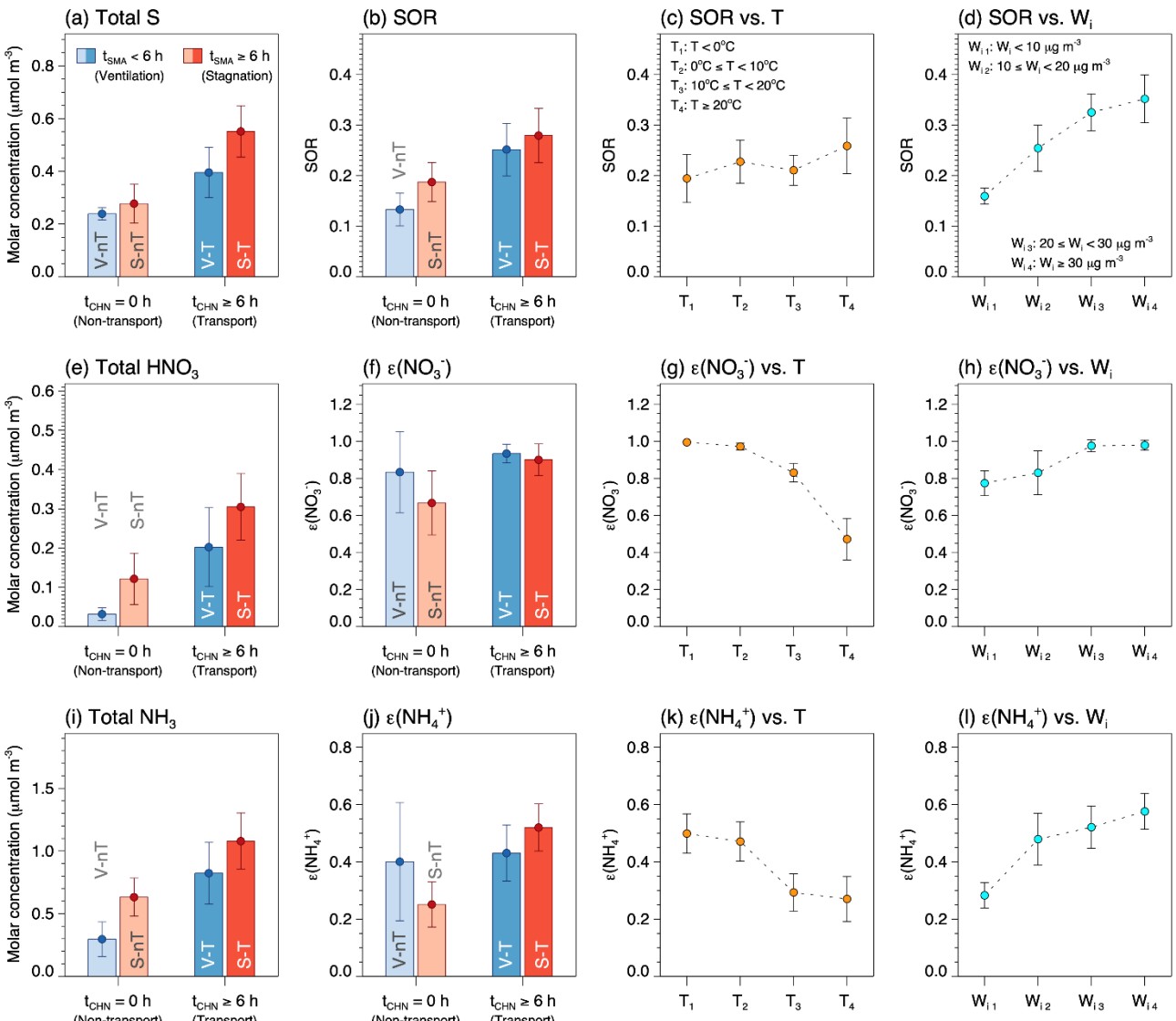

**Figure 10: Average concentrations and confidence intervals at 95% of (a) total sulfur (= SO₂ + SO₄²⁻), (b–d) sulfur oxidation ratio (SOR), (e) total HNO₃ (= HNO₃ + NO₃⁻), (f–h) nitrate partitioning ratio [ε(NO₃⁻)], (i) total NH₃ (= NH₃ + NH₄⁺), and (j–l) ammonium partitioning ratio [ε(NH₄⁺)]. (a–b, e–f, and i–j) Average for the *local ventilation with no regional transport* (V-nT) group (light blue), *local stagnation with no regional transport* (S-nT) group (light red), *local ventilation with regional transport* (V-T) group (darker light blue), and *local stagnation with regional transport* (S-T) group (darker light red). (c, g, and k) Average for the temperature ($T$) ranges of $T < 0°C$ ($T_1$; $n = 19$), $0°C \le T < 10°C$ ($T_2$; $n = 31$), $10°C \le T < 20°C$ ($T_3$; $n = 30$), and $T \ge 20°C$ ($T_4$; $n = 19$). (d, h, and l) Average for the inorganic ALW ($W_i$) ranges of $W_i < 10$ μg m⁻³ ($W_{i1}$; $n = 58$), $10$ μg m⁻³ $\le W_i < 20$ μg m⁻³ ($W_{i2}$; $n = 14$), $20$ μg m⁻³ $\le W_i < 30$ μg m⁻³ ($W_{i3}$; $n = 12$), and $W_i \ge 30$ μg m⁻³ ($W_{i4}$; $n = 15$).**

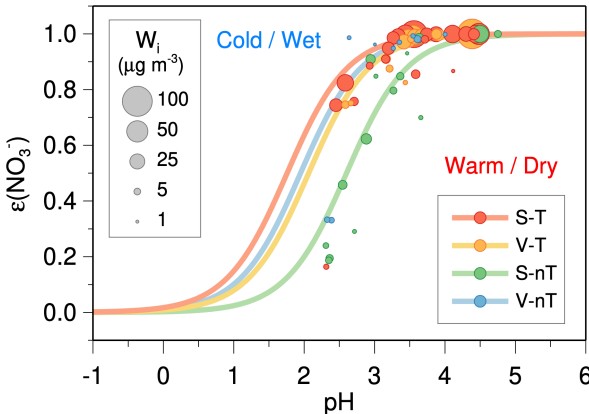

**Figure 11: The calculated sigmoid curves (S-curves) and observed ambient gas-particle partitioning ratios (solid circles) for nitrate [$\varepsilon(NO_3^-)$] plotted against the ISORROPIA-predicted particle pH. The blue, green, yellow, and red circles represent the** *local ventilation with no regional transport* **(V-nT) group,** *local stagnation with no regional transport* **(S-nT) group,** *local ventilation with regional transport* **(V-T) group, and** *local stagnation with regional transport* **(S-T) group, respectively, and the size of each circle is proportional to the concentration of inorganic ALW ($W_i$). The curves were calculated based on the median values of temperature ($T$), $W_i$, and product of activity coefficient ($\gamma_{H^+}\gamma_{NO_3^-}$) for each group; $T = 0.6°C$, $W_i = 2.2$ µg m$^{-3}$, and $\gamma_{H^+}\gamma_{NO_3^-} = 0.146$ for the V-nT group; $T = 17.5°C$, $W_i = 5.6$ µg m$^{-3}$, and $\gamma_{H^+}\gamma_{NO_3^-} = 0.295$ for the S-nT group; $T = 11.6°C$, $W_i = 16.6$ µg m$^{-3}$, and $\gamma_{H^+}\gamma_{NO_3^-} = 0.458$ for the V-T group; $T = 6.9°C$, $W_i = 18.0$ µg m$^{-3}$, and $\gamma_{H^+}\gamma_{NO_3^-} = 0.400$ for the S-T group.**

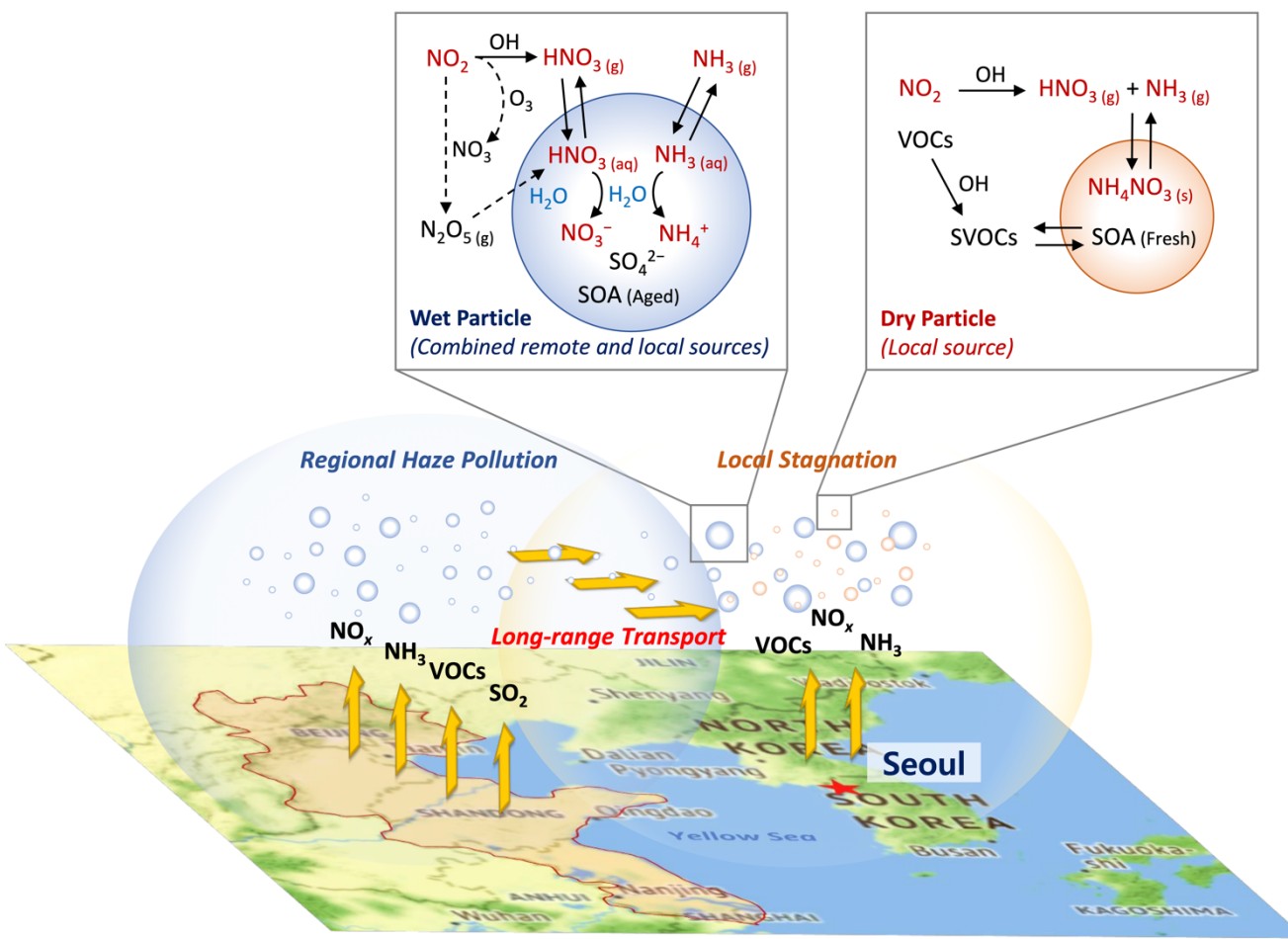

**Figure 12: Schematic of synergistic nitrate partitioning to the particle phase aided by wet particles transported from the remote source area (North China Plain and Yangtze River Delta) in a high NO$_x$ and NH$_3$ urban area (Seoul metropolitan area) during the cold season in East Asia. The background map was derived from © Google Maps.**