# Peer review of "Synergistic enhancement of urban haze by nitrate uptake into transported hygroscopic particles in the Asian continental outflow"

_Atmospheric Chemistry and Physics, 2019_

## Referee Comment (RC1) · Anonymous Referee #2 · 10 Feb 2020

General comments:

Seo et al. combined measurements of PM2.5 mass and composition (from filter collection) in Seoul, Korea, with thermodynamic modeling in ISORROPIA II and back trajectory analysis. They find that particles influenced by regional transport from source areas in China have higher mass, higher inorganic aerosol content and higher water content. Specifically, the highest concentrations are observed in air masses under locally stagnant conditions affected by regional transport. They conclude that the synergistic effects of local stagnation and regional transport affect PM2.5 concentrations and composition.

[Figure]

Except for a few English usage issues, the manuscript is well written and within the scope of Atmospheric Chemistry and Physics. While the measurements are local, the main conclusion(s) are likely applicable to other regions and are therefore of broader interest. I have one major and several minor comments and concerns which should be addressed before publication.

Major comments:

In section 2.2, the author describe reconstructing concentrations of HNO3 and NH3 for their thermodynamic modeling analysis. I have two main issues with this:

1. In my opinion, the current version of the manuscript does not sufficiently justify the methods of reconstruction. For NH3, the authors seem to assume that the concentrations at the Gwangjin site are the same as the concentrations at the KIST site, and that they did not change between years. Please describe why these are reasonable assumptions. For HNO3, the authors seem to assume that the NO3/HNO3 ratio does not depend on the sum of NO3+HNO3 concentrations. Please justify this assumption.

2. The reconstructed concentrations certainly introduce uncertainty, which the authors recognize. For example, they comment (lines 122-123): "Although there are uncertainties in the reconstructed NH3 and HNO3 due to lack of direct measurements, their impact on the estimation of inorganic ALW and particle pH may be small enough." They follow this comment by a discussion on why the impact may be small enough. In my opinion, there is too much uncertainty here (in the data and the language, e.g. "may" and "would"), and I suggest that the authors conduct a sensitivity analysis on how uncertainty in the reconstructed concentrations of NH3 and HNO3 affects their conclusions.

Minor and technical comments:

line 12: replace 'stagnant' with 'stagnation'

line 17: replace 'group' with 'grouped' or otherwise revise as this is unclear

lines 32-35: the sentence is unclear, especially the second half (... " and also a situation..."). Please revise (splitting into two sentences would probably help). Line 86: "The OM identified in this study is ~5% of the total OM." I think I know what you mean, but this sentence is confusing to me. Perhaps rephrase as "The organic compounds identified in this study constitute ~5% of the total OM."

Line 255: replace 'the more increase' with 'the higher increase'?

Lines 291-292: "Interestingly, SOR increase by temperature (and also irradiance) is not significant as much as inorganic ALW (Figs. 8c) despite..." please revise this phrase as it is not clear.

Lines 293-295: "This implies that the observed high SO4 in the S-T group was induced by the aqueous-phase oxidation of SO2 in the transported wet particles rather than the photochemical gas-phase oxidation." It seems appropriate to point out here that gas-phase oxidation likely also played a role (i.e. the data do not rule out gas-phase oxidation as a source of sulfate).

There are several instances where the article "the" is overused. As an example, last sentence in the abstract: "This study reveals the synergistic effect of remote and local sources on the urban haze pollution in the downwind region and provides insight into the nonlinearity of domestic and foreign contributions to receptor PM2.5 concentrations in the numerical air quality models". I would suggest removing "the" in front of 'urban', 'numerical air quality models'. This seems more consistent with common usage and would also further help to suggest applicability of the conclusions to other areas. I suggest the authors review the whole manuscript for use of "the".
* * *

---

## Referee Comment (RC2) · Rodney Weber (Referee) · 4 Mar 2020

This paper assesses the composition of PM2.5 in Seoul under different transport & meteorological conditions. The separation of data collection into periods when Seoul was impacted or not by upwind transport from China, combined with local ventilation conditions over Seoul, is clever and insightful. The result is that clear differences in the aerosol composition and processes can be identified. The authors show that highest PM2.5 concentrations occur when species transported from China are present and when there is little dispersion over Seoul, as might be expected. The unique feature is they find that these periods have enhanced inorganic aerosol concentrations and

investigate a number of possible feedbacks that could explain the enhanced aerosol concentrations under these conditions, all related to aerosol liquid water levels (ALW). This includes enhanced SO2 oxidation to form sulfate and especially the uptake of HNO3 to form particle NO3-. The specific contributions of China (SO2/sulfate) vs Seoul (NOx/NH3) on these interactions is also identified. This allows a unique assessment of possible control strategies to reduce PM2.5 mass. The main issue with this paper is the lack of NH3 and HNO3 data that are required to run the thermodynamic model. The authors should more fully assess this limitation through a detailed sensitivity analysis, but my suspicion is it will not significantly change the result.

Specific Comments.

The use of acronyms made the paper, at times, difficult to follow for me. Where it is possible, it might be better to just write out the term. A list (table) defining them could also be useful. For example, in this study SIA is just sulfate, nitrate and ammonium.

Potential NO3- sampling issues? Were samples gas denuded; seems not but maybe use of Teflon filters minimizes this possible positive artifact? Since particle NO3- is a large component of this paper, and it is known to be difficult to measure using filters due to evaporation, this issue should be discussed. That is, is it possible that ammonium nitrate is significantly under-measuring in this study?

Regarding the estimated NH3 and effect on pH. A sensitivity test is warranted, as noted by another reviewer. This could include discussing epsilon(NH4+) and epsilon(NO3-). Table 2 suggests that with epsilon(NH4+) values ranging from 0.23 to 0.5, there may be some sensitivity to [NH3]. But as noted, this may not contribute to much change in pH. Furthermore given Fig 9, in some situations this may not have a large effect on predicted NO3-. The point is, the epsilon data for NH4+ and NO3- can be used to help assess the sensitivity of the predictions to uncertainties in gas phase species that were not measured; eg, one could make a graph of epsilon(NH4+) similar to Fig 9 and than show the data for a range of estimated NH3 and HNO3 around the predicted values.

Line 319, one could be more specific here, with epsilon(NO3-) near or at 1 there is a direct relation between NOx control and particle NO3-.

Sulfate is a large component of the SIA. How does sulfate play a role in this feedback mechanism (see next)?

The idea of feed back (or sometimes called co-condensation) leading to more uptake of NH3 and HNO3 by the added liquid water is not a new concept. I suggest the authors think about it some more and add a deeper discussion. It happens for any semi-volatile acidic species that when partitioned to the particle phase significantly increases the water uptake, which then raises the pH and allows more uptake. Examples include HCl/Cl- & HNO3/NO3-. Since sulfate is not semi-volatile and highly hygroscopic the semivolatile species involved that is driving this feedback process must generally have significantly higher concentrations then sulfate, or more precisely, contribute comparable or more to AWC than sulfate, otherwise the feedback does not exist. For example, in this study if sulfate was significantly larger the nitrate, would nitrate levels increase due to uptake of water? Probably not because sulfate would then control the overall AWC. One could play around with sulfate concentrations to see when this happens. The process discussed here is very similar to that discussed in Guo et al. (2017). Also, as another example, see Topping et al (2013).

Guo, H., J. Liu, K. D. Froyd, J. Roberts, P. R. Veres, P. L. Hayes, J. L. Jimenez, A. Nenes, and R. J. Weber (2017), Fine particle pH and gas-particle phase partitioning of inorganics in Pasadena, California, during the 2010 CalNex campaign, Atm. Chem. Phys., 17, 5703-5719.

Topping, D., P. Connolly, and G. McFiggans (2013), Cloud droplet number enhanced by co-condensation of organic vapours, Nature Geoscience, 6, 443-446.

---

## Author Comment (AC1) · 15 May 2020

**Response to Referee #2**

**General comments:**

Seo et al. combined measurements of PM$_{2.5}$ mass and composition (from filter collection) in Seoul, Korea, with thermodynamic modeling in ISORROPIA II and back trajectory analysis. They find that particles influenced by regional transport from source areas in China have higher mass, higher inorganic aerosol content and higher water content. Specifically, the highest concentrations are observed in air masses under locally stagnant conditions affected by regional transport. They conclude that the synergistic effects of local stagnation and regional transport affect PM$_{2.5}$ concentrations and composition.

Except for a few English usage issues, the manuscript is well written and within the scope of Atmospheric Chemistry and Physics. While the measurements are local, the main conclusion(s) are likely applicable to other regions and are therefore of broader interest. I have one major and several minor comments and concerns which should be addressed before publication.

We appreciate the reviewer for careful reading and helpful comments that improve our manuscript. As indicated in the following point-by-point responses, we have incorporated the reviewer's comments and suggestions into the revised manuscript. We have conducted additional analyses, modified texts, figures, and tables, and added several new figures and references in the revised version of manuscript. Each response to the reviewer is in blue, and the change in the manuscript is in red.

During the revision process, we found that the molar concentrations of Ca$^{2+}$ and Mg$^{2+}$ in the ISORROPIA input data were mistakenly applied by 2-time values. We have corrected those input data errors and newly conducted simulations. As a result, average and standard deviation of ALW (both $W_i$ and $W_o$), ionic strength, pH, oxidation/partitioning ratios [SOR, $\varepsilon(NO_3^-)$, and $\varepsilon(NH_4^+)$] in Table 2 have been slightly changed. By the changes in results, several figures (Figs. 6a, g, and h, Fig 7, Fig. 8, Fig. 10, and Fig. 11 in the revised version) have also been modified. However, such changes are negligible and did not affect our conclusions.

We also corrected average and standard deviation of NH$_3$ concentrations in Table 2, because the unit of original version were µg m$^{-3}$. NH$_3$ are now represented by unit of ppb in the revised version. Sect. 2.2 and Sect. 2.3 were exchanged with each other to keep consistency of the order of figures and text contents.

**Major comments:**

In section 2.2, the author describes reconstructing concentrations of HNO$_3$ and NH$_3$ for their thermodynamic modeling analysis. I have two main issues with this:

1. In my opinion, the current version of the manuscript does not sufficiently justify the methods of reconstruction. For NH$_3$, the authors seem to assume that the concentrations at the Gwangjin site are the same as the concentrations at the KIST site, and that they did not change between years. Please describe why these are reasonable assumptions.

Since there were no NH$_3$ measurements in Seoul for the analysis period of 2012–2014, we had to reconstruct NH$_3$ concentrations for the thermodynamic model simulations in some way. Fortunately, we found a year-long measurement (2010–2011) of NH$_3$ at the Gwangjin site (37.545°N, 127.096°E, 7.8 km southeastward away from the KIST site) in Seoul reported by Phan et al. (2013). Although the NH$_3$ data reported by Phan et al. (2013) has some spatiotemporal gaps (~1 to 3 years) to data in this study measured at the KIST site (37.603°N, 127.047°E), we assumed that both two site data shares statistical characteristics like annual average NH$_3$ level, standard deviation, and correlations with meteorological factors (temperature, RH, and wind speed) and anthropogenic gas pollutants (SO$_2$, NO$_2$, and CO), as mentioned in Text S1. Our assumption is based on (i) the small interannual changes in the estimated NH$_3$ emissions in Seoul from the Clean Air Policy Support System (CAPSS) inventory (Lee et al., 2011; NIER, 2018) (Fig. S5) and (ii) the similar environment of both two sites that are located downwind of the downtown core in the prevailing westerlies in Seoul and surrounded mainly by residential areas and small urban green areas (modified version of Fig. S1).

To clarify the justification of the $NH_3$ reconstruction based on the data by Phan et al. (2013), we added the following sentences to the end of Line 111 on page 4 of the original version as follows:

There were small interannual changes in the estimated $NH_3$ emission in Seoul (Fig. S5; NIER, 2018), and both two sites share similar environments located downwind from the downtown core under the prevailing westerlies and surrounded by residential and small urban green areas (Fig. S1). Therefore, we assumed that the statistical characteristics of both two sites and periods would be similar despite the temporal gap (~1–3 yr) and spatial distance (~7.8 km) between this study and Phan et al. (2013).

[Figure]

Figure S1: (a) The locations of Seoul (left panel) and the Korea Institute of Science and Technology (KIST) $PM_{2.5}$ sampling site (red triangle), the Korea Meteorological Administration (KMA) weather station (blue diamond), and Korea Ministry of Environment (KMOE) air quality monitoring sites (34 solid circles in yellow and green) in Seoul (right panel). The solid circle in green indicates the Gwangjin site at which a year-long $NH_3$ measurement was conducted by Phan et al. (2013). (b–c) Satellite maps of (b) the KIST $PM_{2.5}$ sampling site (37.603°N, 127.047°E) utilized in this study and (c) the Gwangjin site (37.545°N, 127.096°E) for $NH_3$ measurement in Phan et al. (2013). Yellow arrows are the sampling locations, areas shaded by light green are urban green areas, and lines in orange are the main roads. The background satellite images are courtesy of Google Earth.

[Figure]

Figure S5: Estimated annual NH₃ emissions in Seoul for 2008–2015 from the Clean Air Policy Support System (CAPSS) inventory (NIER, 2018).

Lee, D.-G., Lee, Y.-M., Jang, K., Yoo, C., Kang, K., Lee, J.- H., Jung, S., Park, J., Lee, S.-B., Han, J., Hong, J., and Lee, S.: Korean national emissions inventory system and 2007 air pollutant emissions, Asian J. Atmos. Environ., 5, 278–291, https://doi.org/10.5572/ajae.2011.5.4.278, 2011.

NIER (National Institute of Environmental Research): National air pollutants emission 2015 (NIER-GP2017-210), NIER, Incheon, South Korea, available at: http://webbook.me.go.kr/DLi-File/NIER/09/023/5668670.pdf (last access: 21 October 2019), 2018 (in Korean).

For HNO₃, the authors seem to assume that the NO₃/HNO₃ ratio does not depend on the sum of NO₃+HNO₃ concentrations. Please justify this assumption.

In general, the $NO_3^-$ / HNO₃ ratio depends on temperature, aerosol liquid water (ALW), and pH. Among these three variables, temperature is an independent meteorological variable, and ALW is dependent on the sum of salts and relative humidity. Therefore, the $NO_3^-$ / HNO₃ ratio obtained from the simulation with reconstructed NH₃ (Simulation 2 in Fig. 4 in the revised manuscript) can be changed solely by the potential changes in pH owing to the increase in total HNO₃ (from $NO_3^-$ to $NO_3^-$ + HNO₃). In the NH₃-rich condition like Seoul, the potential changes in pH can be small due to the buffering effect of NH₃ partitioning (Weber et al., 2016). Therefore, we can use the $NO_3^-$ / HNO₃ ratio from Simulation 2 and the measured $NO_3^-$ to estimate the total HNO₃. Fig. 4 shows comparisons of pH, $\varepsilon(NO_3^-)$, and $\varepsilon(NH_4^+)$ among three simulations (Simulation 1 with only ions, Simulation 2 with ions and reconstructed NH₃, and Simulation 3 with ions, reconstructed NH₃, and HNO₃ estimated by the $NO_3^-$ / HNO₃ ratio from Simulation 2), and here we can see that the pH is largely changed by the introduction of NH₃ gas but is nearly not changed by the increase in total HNO₃.

To justify the estimation of HNO₃ by using the $NO_3^-$ / HNO₃ ratio, we added the following sentences to Line 119 on page 4 of the original version as follows:

Note that we regarded the HNO₃ / $NO_3^-$ ratio as approximately independent of the total HNO₃. Although the nitrate partitioning depends on temperature, ALW content, and pH in general, the HNO₃ / $NO_3^-$ ratio is affected alone by pH because both temperature and ALW are independent variables in this step. In the existence of excess NH₃ as we assumed for Seoul (~10 ppb at the first step), changes in pH by total HNO₃ are limited owing to the buffering effect of NH₃–$NH_4^+$ partitioning (Weber et al., 2016; Guo et al., 2018; Lim et al., 2020). The pH, nitrate partitioning ratio [$\varepsilon(NO_3^-)$ = $NO_3^-$ / (HNO₃ + $NO_3^-$)], and ammonium partitioning ratio [$\varepsilon(NH_4^+)$ = $NH_4^+$ / ($NH_3$ + $NH_4^+$)] from the ISORROPIA simulations with and without gas-phase HNO₃ information show negligible differences between each other (Figs. 4d–f), unlike the significant role of additional NH₃ information on increasing pH and $\varepsilon(NO_3^-)$ and decreasing $\varepsilon(NH_4^+)$ (Figs. 4a–c).

Also, we added Fig. 4 into the manuscript in the revised version.

[Figure]

Figure 4: Comparisons of the predicted pH, nitrate partitioning ratio [$\varepsilon(NO_3^-) = NO_3^- / (HNO_3 + NO_3^-)$], and ammonium partitioning ratio [$\varepsilon(NH_4^+) = NH_4^+ / (NH_3 + NH_4^+)$]] between (a–c) the simulation without gas-phase $NH_3$ and $HNO_3$ information (using $NH_4^+$ and $NO_3^-$ as the total $NH_3$ and total $HNO_3$; Simulation 1) and the simulation with reconstructed $NH_3$ (using $NH_3 + NH_4^+$ as the total $NH_3$ but only $NO_3^-$ as the total $HNO_3$; Simulation 2) and (d–f) the Simulation 2 and the simulation with both estimated gas-phase $NH_3$ and $HNO_3$ information (this study; Simulation 3). Filled circles in light blue, light green, light yellow, and light red colors represent daily data belong to the *local ventilation with no regional transport* (V-nT) group, *local stagnation with no regional transport* (S-nT) group, *local ventilation with regional transport* (V-T) group, and *local stagnation with regional transport* (S-T) group, respectively. Gray solid lines indicate a 1-to-1 relationship.

2. The reconstructed concentrations certainly introduce uncertainty, which the authors recognize. For example, they comment (lines 122-123): "Although there are uncertainties in the reconstructed $NH_3$ and $HNO_3$ due to lack of direct measurements, their impact on the estimation of inorganic ALW and particle pH may be small enough." They follow this comment by a discussion on why the impact may be small enough. In my opinion, there is too much uncertainty here (in the data and the language, e.g. "may" and "would"), and I suggest that the authors conduct a sensitivity analysis on how uncertainty in the reconstructed concentrations of $NH_3$ and $HNO_3$ affects their conclusions.

As the reviewer suggested, we conducted sensitivity tests to examine how various $NH_3$ levels can affect and change the results, particularly pH and inorganic partitioning ratios [$\varepsilon(NO_3^-)$ and $\varepsilon(NH_4^+)$]. Various $NH_3$ levels from 0 ppb to 50 ppb are applied for all the measurement days as fixed values (no day-to-day variations in $NH_3$).

In the reconstructed $NH_3$ data contains two types of error that are (i) from the day-to-day estimation by the multiple regression model (Text S1 and Table S1) and (ii) from the average concentration of $NH_3$ in Seoul. Figs. 5a, c, and e show good correlations between the results from simulation with daily varied $NH_3$ (average of 10.9 ppb) and that with daily fixed $NH_3$ [e.g., $R^2$ values of 0.779 for pH, 0.984 for $\varepsilon(NO_3^-)$, and 0.575 for $\varepsilon(NH_4^+)$ in 10 ppb]. If the fixed $NH_3$ level was not 0 ppb, correlations with 5, 10, 20, and 40 ppb are not significantly different. Therefore, the potential errors in pH and partitioning ratios raised from the day-to-day estimation seem to be limited. Figs. 5b, d, and f show that the average pH and partitioning ratios for each categorized group become flattened over the 5 ppb of $NH_3$. This can be already expected from the buffering effect of $NH_3$–$NH_4^+$

partitioning on pH, and thus the potential errors from the estimation of NH₃ do not change our conclusion.

Based on the sensitivity test, we removed words representing uncertainty from this section, and we added the content related to the sensitivity test for NH₃ to Line 130 on page 5 of the original manuscript as follows:

We conducted ISORROPIA simulations with various NH₃ levels (0–50 ppb) applied for all the measurement days as fixed values to further explore the sensitivity of pH, $\varepsilon(NO_3^-)$, and $\varepsilon(NH_4^+)$ to NH₃. Good correlations between results from the daily varied NH₃-simulation and the fixed NH₃-simulations shows that potential errors induced by the estimation of daily NH₃ levels will be small if NH₃ concentrations are nonzero (Figs. 5a, c, and e). Increasing pH and $\varepsilon(NO_3^-)$ and decreasing $\varepsilon(NH_4^+)$ by the increase in fixed NH₃ level become flatten over the range from a half to 2 times of the average NH₃ concentration in Seoul (~5–20 ppb; Phan et al., 2013) (Figs. 5b, d, and f), and this indicates that the potential errors in reconstructed NH₃ will not significantly change the results in this study.

Also, we added Fig. 5 into the manuscript in the revised version.

[Figure]

Figure 5: Sensitivity of (a–b) predicted pH, (c–d) nitrate partitioning ratio [$\varepsilon(NO_3^-) = NO_3^- / (HNO_3 + NO_3^-)$], and (e–f) ammonium partitioning ratio [$\varepsilon(NH_4^+) = NH_4^+ / (NH_3 + NH_4^+)$] to gas-phase ammonia (NH₃) concentrations. (a, c, and e) Comparisons between the simulation in this study (using the daily reconstructed NH₃ concentrations) and the simulations with the various fixed NH₃ levels (colored dots in gray, blue, green, yellow, and red for 0 ppb, 5 ppb, 10 ppb, 20 ppb, and 40 ppb, respectively). Values in round brackets are the R-squared values of the linear regressions. Gray solid lines indicate a 1-to-1 relationship. (b, d, and f) Average (solid circles) and standard deviation (vertical bars) of pH, $\varepsilon(NO_3^-)$, and $\varepsilon(NH_4^+)$ for the *local ventilation with no regional transport* (V-nT) group (light blue), *local stagnation with no regional transport* (S-nT) group (light green), *local ventilation with regional transport* (V-T) group (light yellow), and *local stagnation with regional transport* (S-T) group (light red) with respect to the various fixed NH₃ levels from 0 ppb to 50 ppb.

Averages and standard deviations of the reconstructed NH₃ concentrations and obtained pH, $\varepsilon(NO_3^-)$, and $\varepsilon(NH_4^+)$ for the V-nT, S-nT, V-T, and S-T groups are represented as blue, green, yellow, and red diamonds with horizontal and vertical bars.

**Minor and technical comments:**

Line 12: replace 'stagnant' with 'stagnation'

Thanks for the correction. It was now corrected.

Line 17: replace 'group' with 'grouped' or otherwise revise as this is unclear

We revised the sentence as follows:

PM$_{2.5}$ measured under the condition of regional transport from the upwind source areas in China was higher in mass concentration and richer in secondary inorganic aerosol (SIA) species and aerosol liquid water (ALW) compared to that measured under the non-transport condition.

Lines 32-35: the sentence is unclear, especially the second half (. . . " and also a situation. . ."). Please revise (splitting into two sentences would probably help).

As the reviewer suggested, we split the sentence into two as follows:

Synoptic weather conditions cause stagnation and long-range transport that can lead to the accumulation of particles and gaseous precursors from local and remote sources. It can also change local meteorological factors to favorable conditions for secondary aerosol production, such as photo-oxidation and aqueous-phase processing (Sun et al., 2014; Zheng et al., 2015; Seo et al., 2017).

Line 86: "The OM identified in this study is ～5% of the total OM." I think I know what you mean, but this sentence is confusing to me. Perhaps rephrase as "The organic compounds identified in this study constitute ～5% of the total OM."

The sentence was now revised as suggested by the reviewer.

Line 255: replace 'the more increase' with 'the higher increase'?

'the more increase' is now replaced with 'the higher increase.'

Lines 291-292: "Interestingly, SOR increase by temperature (and also irradiance) is not significant as much as inorganic ALW (Figs. 8c) despite. . ." please revise this phrase as it is not clear.

We revise the sentence as follows:

Interestingly, the increasing SOR by temperature (and also irradiance) is not significant as much as that by inorganic ALW (Figs. 8c) despite the high-temperature and strong-irradiance conditions conducive to photochemical oxidation of SO$_2$ in summer.

Lines 293-295: "This implies that the observed high SO$_4$ in the S-T group was induced by the aqueous-phase oxidation of SO$_2$ in the transported wet particles rather than the photochemical gas-phase oxidation." It seems appropriate to point out here that gas-phase oxidation likely also played a role (i.e. the data do not rule out gas-phase oxidation as a source of sulfate).

As the reviewer suggested, we modified the sentence not to rule out gas-phase oxidation as follows:

This implies that the observed high SO$_4^{2-}$ in the S-T group resulted not only from the photochemical gas-phase oxidation but also considerably from the aqueous-phase oxidation of SO$_2$ in the transported wet particles.

There are several instances where the article "the" is overused. As an example, last sentence in the abstract: "This study reveals the synergistic effect of remote and local sources on the urban haze pollution in the downwind region and provides insight into the nonlinearity of domestic and foreign contributions to receptor

PM$_{2.5}$ concentrations in the numerical air quality models". I would suggest removing "the" in front of 'urban', 'numerical air quality models'. This seems more consistent with common usage and would also further help to suggest applicability of the conclusions to other areas. I suggest the authors review the whole manuscript for use of "the".

Thank you for comments. We removed several "the" in the revised version of manuscript following the reviewer's suggestion.

---

## Author Comment (AC2) · 15 May 2020

**Response to Referee #1 (Dr. Rodney Weber)**

This paper assesses the composition of $PM_{2.5}$ in Seoul under different transport & meteorological conditions. The separation of data collection into periods when Seoul was impacted or not by upwind transport from China, combined with local ventilation conditions over Seoul, is clever and insightful. The result is that clear differences in the aerosol composition and processes can be identified. The authors show that highest $PM_{2.5}$ concentrations occur when species transported from China are present and when there is little dispersion over Seoul, as might be expected. The unique feature is they find that these periods have enhanced inorganic aerosol concentrations and investigate a number of possible feedbacks that could explain the enhanced aerosol concentrations under these conditions, all related to aerosol liquid water levels (ALW). This includes enhanced $SO_2$ oxidation to form sulfate and especially the uptake of $HNO_3$ to form particle $NO_3^-$. The specific contributions of China ($SO_2$ / sulfate) vs Seoul ($NO_x$ / $NH_3$) on these interactions is also identified. This allows a unique assessment of possible control strategies to reduce $PM_{2.5}$ mass. The main issue with this paper is the lack of $NH_3$ and $HNO_3$ data that are required to run the thermodynamic model. The authors should more fully assess this limitation through a detailed sensitivity analysis, but my suspicion is it will not significantly change the result.

We appreciate the reviewer for valuable comments and suggestions. As indicated in the following point-by-point responses, we have incorporated the reviewer's comments and suggestions into the revised manuscript. We have conducted additional analyses, modified texts, figures, and tables, and added several new figures (Figs. 4 and 5) and references in the revised version. Each response to the reviewer is in blue, and the change in the manuscript is in red.

During the revision process, we found that the molar concentrations of $Ca^{2+}$ and $Mg^{2+}$ in the ISORROPIA input data were mistakenly applied by 2-time values. We have corrected those input data errors and newly conducted simulations. As a result, average and standard deviation of ALW (both $W_i$ and $W_o$), ionic strength, pH, oxidation/partitioning ratios [SOR, $\varepsilon(NO_3^-)$, and $\varepsilon(NH_4^+)$] in Table 2 have been slightly changed. By the changes in results, several figures (Figs. 6a, g, and h, Fig 7, Fig. 8, Fig. 10, and Fig. 11 in the revised version) have also been modified. However, such changes are negligible and did not affect our conclusions.

We also corrected average and standard deviation of $NH_3$ concentrations in Table 2, because the unit of original version were $\mu g\ m^{-3}$. $NH_3$ are now represented by unit of ppb in the revised version. Sect. 2.2 and Sect. 2.3 were exchanged with each other to keep consistency of the order of figures and text contents.

**Specific Comments:**

The use of acronyms made the paper, at times, difficult to follow for me. Where it is possible, it might be better to just write out the term. A list (table) defining them could also be useful. For example, in this study SIA is just sulfate, nitrate and ammonium.

Thanks for the comment. We tried to reduce the usage of acronyms in the revised version.

Potential $NO_3^-$ sampling issues? Were samples gas denuded; seems not but maybe use of Teflon filters minimizes this possible positive artifact? Since particle $NO_3^-$ is a large component of this paper, and it is known to be difficult to measure using filters due to evaporation, this issue should be discussed. That is, is it possible that ammonium nitrate is significantly under-measuring in this study?

As the reviewer pointed out, we did not employ both a denuder and backup filters for $PM_{2.5}$ sampling on Teflon filters. Therefore, there are possibilities of both positive and negative artifacts by absorption of gas-phase nitric acid on the filter and evaporation of ammonium nitrate. Although both artifacts may partly compensate each other, previous evaluation studies showed that the major concern of nitrate measurement on Teflon filter is the negative artifact (Ashbaugh and Eldred, 2004; Chow et al., 2005). Nie et al. (2010) reported large nitrate loss (~75%) at the lower nitrate concentrations (< 10 $\mu g\ m^{-3}$) but much smaller nitrate loss (~10%) at the higher

nitrate concentrations (> 10 µg m$^{-3}$) attributable to formation of particle cake, even in summertime (temperature range of 22–34°C) in Beijing. Considering that the evaporative loss of nitrate was minimal in winter, they expected small to moderate sampling artifacts for conventional sampling method (un-denuded filter sampling without backup filters) in polluted eastern China.

In this study, average temperature of two *regional transport* (V-T and S-T) groups are much lower than summer (~9°C), and average nitrate concentrations of these two groups are high (> 10 µg m$^{-3}$). In addition, although average nitrate concentrations of the *local ventilation with no regional transport* (V-nT) group are very low (~2 µg m$^{-3}$), the cold temperature (~3°C) can prevent evaporation of ammonium nitrate from the Teflon filter. Therefore, we expected potential nitrate loss for these three groups to be small enough (< 20%). The *local stagnation with no regional transport* (S-nT) group is probably most affected by evaporation because of its low nitrate concentration (~5 µg m$^{-3}$) with moderate temperature (~15°C). However, nitrate loss of this group would not be as much as that of the summertime Beijing (~75%) reported by Nie et al. (2010).

We further examined the sensitivity of our results by potential negative artifacts in NO$_3^-$ through the ISORROPIA simulations using hypothetical NH$_4^+$ and NO$_3^-$ concentrations considering evaporative nitrate loss [from 0% (no loss) to 80% loss of ammonium nitrate from the filter samples] together with the average concentrations of PM$_{2.5}$ components for each categorized group summarized in Table 2 (Fig. S8). Note that the results from the 0% nitrate loss assumption do not exactly same as the values in Table 2, because Table 2 shows the average of measured and predicted data for each group while Fig. S8 is obtained from the simulations using the average inorganic concentrations. If we assumed a 20% nitrate loss [NO$_3^-$ and excess NH$_4^+$ = (NH$_4^+$ / SO$_4^{2-}$ − 1.5) × SO$_4^{2-}$; Pathak et al., 2004] for the V-nT, V-T, and S-T groups and a 50% nitrate loss for the S-nT group based on above discussion, the nitrate fraction in PM$_{2.5\ dry}$ of the S-nT group becomes comparable to that of the *regional transport* (V-T and S-T) groups (Fig. S8a). However, inorganic ALW ($W_i$), $\varepsilon$(NO$_3^-$), and $\varepsilon$(NH$_4^+$) of the V-T and S-T groups are still larger than that of the S-nT group (Figs. S8b, d, and e), and this supports an important role of the transported wet particles in the formation of PM$_{2.5}$ inorganic species. In conclusion, therefore, potential negative artifacts induced by the un-denuded filter sampling method without backup filters do not significantly change the conclusions of this study.

To clarify the effect of potential errors from the sampling artifacts on our results, we added the following paragraph to the end of Line 90 on page 3 (as the third paragraph in Sect. 2.1).

Note that the PM$_{2.5}$ sampling on Teflon filter for inorganic ions was conducted without both a denuder and backup filters, and thus there could be potential sampling artifacts on the results, particularly negative artifacts in semivolatile ammonium nitrate (Ashbaugh and Eldred, 2004; Chow et al., 2005). Nie et al. (2010) reported that summertime nitrate loss on Teflon filter from the un-denuded filter sampling without backup filters is to be ~75% at lower nitrate concentrations (< 10 µg m$^{-3}$) but only ~10% at higher nitrate concentrations (> 10 µg m$^{-3}$) due to the formation of particle cake. Considering small evaporative loss in the cold season and the high nitrate concentration in Seoul, we expected small to moderate sampling errors in this study. Sensitivity tests considering potential ammonium nitrate loss from the filter samples show that the assumption of 20% nitrate loss for the high concentrations with low-temperature groups and 50% nitrate loss for the low concentrations with moderate temperature group does not change our conclusion (Fig. S8).

[Figure]

Figure S8: Sensitivity of (a) nitrate fraction in dry PM$_{2.5}$ (NO$_3^-$ / PM$_{2.5\ dry}$), (b) nitrate partitioning ratio [$\varepsilon$(NO$_3^-$)], (c) ammonium fraction in dry PM$_{2.5}$ (NH$_4^+$ / PM$_{2.5}$), (d) ammonium partitioning ratio [$\varepsilon$(NH$_4^+$)], (e) inorganic ALW ($W_i$) content, and (f) pH to the hypothetical ammonium nitrate loss during the sampling on Teflon filters. Average PM$_{2.5}$ components (Table 2) with extrapolated concentrations of NO$_3^-$ and excess NH$_4^+$ [NH$_4^+{}_{excess}$ = (NH$_4^+$ / SO$_4^{2-}$ − 1.5) × SO$_4^{2-}$; Pathak et al., 2004], considering the hypothetical ammonium nitrate loss from 0% to 80%, were employed in the ISORROPIA simulations. Sensitivity curves in blue, green, yellow, and red colors represent the *local ventilation with no regional transport* (V-nT) group, *local stagnation with no regional transport* (S-nT) group, *local ventilation with regional transport* (V-T) group, and *local stagnation with regional transport* (S-T) group, respectively.

Ashbaugh, L. L. and Eldred R. A.: Loss of particle nitrate from Teflon sampling filters: Effects on measured gravimetric mass in California and in the IMPROVE network, J. Air Waste Manage. Assoc., 54, 93–104, https://doi.org/10.1080/10473289.2004.10470878, 2004.

Chow, J. C., Watson, J. G., Lowenthal, D. H., and Magliano, K. L.: Loss of PM$_{2.5}$ nitrate from filter samples in central California, J. Air Waste Manage. Assoc., 55, 1158–1168, https://doi.org/10.1080/10473289.2005.10464704, 2005.

Nie, W., Wang, T., Gao, X., Pathak, R. K., Wang, X., Gao, R., Zhang, Q., Yang, L., and Wang, W.: Comparison among filter-based, impactor-based and continuous techniques for measuring atmospheric fine sulfate and nitrate, Atmos. Environ., 44, 4396–4403, https://doi.org/10.1016/j.atmosenv.2010.07.047, 2010.

Pathak, R. K., Yao, X., and Chan, C. K.: Sampling artifacts of acidity and ionic species in PM$_{2.5}$, Environ. Sci. Technol., 38, 254–259, https://doi.org/10.1021/es0342244, 2004.

Regarding the estimated $NH_3$ and effect on pH. A sensitivity test is warranted, as noted by another reviewer. This could include discussing $\varepsilon(NH_4^+)$ and $\varepsilon(NO_3^-)$. Table 2 suggests that with $\varepsilon(NH_4^+)$ values ranging from 0.23 to 0.5, there may be some sensitivity to $[NH_3]$. But as noted, this may not contribute to much change in pH. Furthermore given Fig 9, in some situations this may not have a large effect on predicted $NO_3^-$. The point is, the epsilon data for $NH_4^+$ and $NO_3^-$ can be used to help assess the sensitivity of the predictions to uncertainties in gas phase species that were not measured; eg, one could make a graph of $\varepsilon(NH_4^+)$ similar to Fig 9 and then show the data for a range of estimated $NH_3$ and $HNO_3$ around the predicted values.

Following the reviewer's suggestion, we conducted sensitivity tests for pH and inorganic partitioning ratios $[\varepsilon(NO_3^-)$ and $\varepsilon(NH_4^+)]$ and represented the results as Fig. 5 in the revised version. As the reviewer expected, the changes in pH, $\varepsilon(NO_3^-)$, and $\varepsilon(NH_4^+)$ by $NH_3$ is not significant when $NH_3$ concentration is larger than 5 ppb.

Various $NH_3$ levels from 0 ppb to 50 ppb are applied for all the measurement days as fixed values (no day-to-day variations in $NH_3$) in our sensitivity tests. Figs. 5a, c, and e show good correlations between the results from simulation with daily varied $NH_3$ (average of 10.9 ppb) and that with daily fixed $NH_3$ if it was not 0 ppb. This indicates that the potential errors in pH and partitioning ratios from the day-to-day estimation seem to be limited. Figs. 5b, d, and f show that the average pH and partitioning ratios for each categorized group become flattened over the 5 ppb of $NH_3$. Therefore, the potential errors from the estimation of $NH_3$ do not change our conclusion.

We added the content related to the sensitivity test for $NH_3$ to Line 130 on page 5 of the original manuscript as follows:

We conducted ISORROPIA simulations with various $NH_3$ levels (0–50 ppb) applied for all the measurement days as fixed values to further explore the sensitivity of pH, $\varepsilon(NO_3^-)$, and $\varepsilon(NH_4^+)$ to $NH_3$. Good correlations between results from the daily varied $NH_3$-simulation and the fixed $NH_3$-simulations shows that potential errors induced by the estimation of daily $NH_3$ levels will be small if $NH_3$ concentrations are nonzero (Figs. 5a, c, and e). Increasing pH and $\varepsilon(NO_3^-)$ and decreasing $\varepsilon(NH_4^+)$ by the increase in fixed $NH_3$ level become flatten over the range from a half to 2 times of the average $NH_3$ concentration in Seoul (~5–20 ppb; Phan et al., 2013) (Figs. 5b, d, and f), and this indicates that the potential errors in reconstructed $NH_3$ will not significantly change the results in this study.

[Figure]

Figure 5: Sensitivity of (a–b) predicted pH, (c–d) nitrate partitioning ratio [$\varepsilon(NO_3^-) = NO_3^- / (HNO_3 + NO_3^-)$], and (e–f) ammonium partitioning ratio [$\varepsilon(NH_4^+) = NH_4^+ / (NH_3 + NH_4^+)$] to gas-phase ammonia ($NH_3$) concentrations. (a, c, and e) Comparisons between the simulation in this study (using the daily reconstructed $NH_3$ concentrations) and the simulations with the various fixed $NH_3$ levels (colored dots in gray, blue, green, yellow, and red for 0 ppb, 5 ppb, 10 ppb, 20 ppb, and 40 ppb, respectively). Values in round brackets are the R-squared values of the linear regressions. Gray solid lines indicate a 1-to-1 relationship. (b, d, and f) Average (solid circles) and standard deviation (vertical bars) of pH, $\varepsilon(NO_3^-)$, and $\varepsilon(NH_4^+)$ for the *local ventilation with no regional transport* (V-nT) group (light blue), *local stagnation with no regional transport* (S-nT) group (light green), *local ventilation with regional transport* (V-T) group (light yellow), and *local stagnation with regional transport* (S-T) group (light red) with respect to the various fixed $NH_3$ levels from 0 ppb to 50 ppb. Averages and standard deviations of the reconstructed $NH_3$ concentrations and obtained pH, $\varepsilon(NO_3^-)$, and $\varepsilon(NH_4^+)$ for the V-nT, S-nT, V-T, and S-T groups are represented as blue, green, yellow, and red diamonds with horizontal and vertical bars.

Line 319, one could be more specific here, with $\varepsilon(NO_3^-)$ near or at 1 there is a direct relation between $NO_x$ control and particle $NO_3^-$.

We added following sentence behind the line 319:

Such a direct relationship between $NO_x$ control and nitrate aerosol is significant at the condition of $\varepsilon(NO_3^-)$ close to 1.

Sulfate is a large component of the SIA. How does sulfate play a role in this feedback mechanism (see next)?

The idea of feedback (or sometimes called co-condensation) leading to more uptake of $NH_3$ and $HNO_3$ by the added liquid water is not a new concept. I suggest the authors think about it some more and add a deeper discussion. It happens for any semi-volatile acidic species that when partitioned to the particle phase

significantly increases the water uptake, which then raises the pH and allows more uptake. Examples include $HCl / Cl^-$ & $HNO_3 / NO_3^-$. Since sulfate is not semi-volatile and highly hygroscopic the semivolatile species involved that is driving this feedback process must generally have significantly higher concentrations then sulfate, or more precisely, contribute comparable or more to AWC than sulfate, otherwise the feedback does not exist. For example, in this study if sulfate was significantly larger then nitrate, would nitrate levels increase due to uptake of water? Probably not because sulfate would then control the overall AWC. One could play around with sulfate concentrations to see when this happens. The process discussed here is very similar to that discussed in Guo et al. (2017). Also, as another example, see Topping et al (2013).

Guo, H., J. Liu, K. D. Froyd, J. Roberts, P. R. Veres, P. L. Hayes, J. L. Jimenez, A. Nenes, and R. J. Weber (2017), Fine particle pH and gas-particle phase partitioning of inorganics in Pasadena, California, during the 2010 CalNex campaign, Atm. Chem. Phys., 17, 5703–5719.

Topping, D., P. Connolly, and G. McFiggans (2013), Cloud droplet number enhanced by co-condensation of organic vapours, Nature Geoscience, 6, 443–446.

We appreciate the reviewer's comment and suggestion on this. As the reviewer pointed out, the significant nitrate concentration compared to sulfate can promote the feedback mechanism including water and nitrate uptake into the particle together with pH increase. This relationship can be also found in our results. For example, the nitrate-to-sulfate molar ratio is largest in the *local stagnation with regional transport* (S-T) group (1.87), followed by the *local ventilation with regional transport* (V-T) group (1.60), the *local stagnation with no regional transport* (S-nT) group (1.29), and the *local ventilation with no regional transport* (V-nT) group (0.81). Because inorganic ALW, pH, and $\varepsilon(NO_3^-)$ are high in the *regional transport* (V-T and S-T) groups followed by the S-nT and V-nT groups (Table 2), this feedback process explains the synergistic effect of transported (wet) particle on the nitrate uptake.

We added the content related to the feedback mechanism to Line 285 on page 9 of the original manuscript as follows:

In terms of the synergistic increase in $NO_3^-$ with ALW, the ratio between $NO_3^-$ and $SO_4^{2-}$ can be an important factor. Hygroscopic uptake of ALW by both $SO_4^{2-}$ and $NO_3^-$ can increase pH by dilution effect on hydrogen ions. Because $NO_3^-$ is a semi-volatile hygroscopic species, the higher pH increased by ALW allows more partitioning of $HNO_3$ gas into the particle phase, and uptake more ALW. However, if $SO_4^{2-}$ is dominant in the particle, such a feedback process will be weakened because sulfate is non-volatile (Guo et al., 2017). The average nitrate-to-sulfate molar ratios of the *regional transport* groups (1.87 for the S-T group and 1.60 for the V-T group) are higher than that of the no regional transport groups (1.29 for the S-nT group and 0.81 for the V-nT group). Since ALW, pH, and $\varepsilon(NO_3^-)$ in the *regional transport* (V-T and S-T) groups are higher than those in the *no regional transport* (V-nT and S-nT) groups, this feedback process can explain the synergistic effect of transported particle on high $NO_3^-$ and ALW fractions.

---

## Author Response (AR2)

**Response to Editor**

We appreciate the comments and corrections from the editor. We have corrected words and sentences, as indicated by the editor's comments. Each response is in blue, and the change in the manuscript is in red.

l. 99: Also figures in the supplement should be referred to in order. Thus, Figure S8 should not be cited right after Figure S2. Please renumber the figures accordingly or change when they are referred to in the text.

Fig. S8 in the manuscript was now Fig. S3 in the revised version. Accordingly, Figs. S3–S7 in the manuscript were also corrected into Figs. S4–S8 in the revised version. The order of the figures in supplement is also changed according to the revised manuscript.

l. 121: (0 h days) - Is this a typo or is there anything missing? Please clarify.

To clarify, 'the medians of both $t_{SMA}$ and $t_{CHN}$ (for $t_{CHN} \neq 0$ h days) were ~6 h' was now replaced by 'the the medians of both $t_{SMA}$ (for all measurement days) and $t_{CHN}$ (for the measurement days with $t_{CHN} \neq 0$ h) were ~6 h' in the revised manuscript.

l. 186: Specify 'small enough' (for what or for which assumptions?)

We added words to the end of the sentence as follows:

Although there are uncertainties in the reconstructed $NH_3$ and $HNO_3$ due to lack of direct measurements, their impact on the estimation of inorganic ALW and particle pH is small enough to utilize them for the thermodynamic analysis in this study.

l. 197 – 198: 'Increasing pH and $\varepsilon(NO_3^-)$ and decreasing $\varepsilon(NH_4^+)$ by the increase in fixed $NH_3$ level become flatten over the 5 ppb of $NH_3$ (Figs. 5b, d, and f), …'

This sentence does not read well and thus the meaning is not clear. Clarify and reword.

In the revised version, we modified the sentence as follows:

The slopes of pH, $\varepsilon(NO_3^-)$, and $\varepsilon(NH_4^+)$ curves to the increase in fixed $NH_3$ level become gradually flat at over 5 ppb (Figs. 5b, d, and f), …

l. 302: SOR only refers to the sulfur oxidation ratio. Is this meant here or do refer by 'higher oxidation ratio' also to the higher fraction of nitrate that is formed from precursors $NO_x$? I suggest either simply adding 'sulfur' to 'ratios of sulfur oxidation' or clarifying that $NO_x$//nitrate is included, respectively.

As the editor suggested, we modified it as 'the higher ratios of sulfur oxidation (SOR) and partitioning.'

l. 326: replace 'particle' by 'particles'

It was now corrected.

l. 331: replace 'colder temperature' by 'lower temperature'

It was now corrected.

l. 332: replace 'help' by 'lead to' or 'cause'

'help' was replaced by 'lead to' in the revised version.

l. 343: Usually the units of the effective Henry's law constant $H^*$ are the same as for the physical Henry's law constant (e.g. M / atm or mol / kg / atm) as $H^* = H (1 + K_a / [H^+])$ whereas $K_a$ is the acid dissociation constant. Thus, the units are not clear unless you define $K_a$ differently. Maybe all what is needed is to add correct units to $K_a$ (or to $H^*$, respectively).

Here we calculated the effective Henry's law constant based on the following expression (Eq. 15) in Clegg et al. (1998).

$$\ln[^x K_\mathrm{H}(\mathrm{HNO_3})] = 385.972199 - 3020.3522\,/\,T - 71.001998\,\ln(T) + 0.131442311\,T - 0.420928363 \times 10^{-4}\,T^2$$

where $^x K_\mathrm{H}(\mathrm{HNO_3})$ is the molality-based effective Henry's law constant of $\mathrm{HNO_3}$ and $T$ is temperature. This approximate expression has been also used in previous thermodynamic model studies (e.g., Guo et al., 2018). Therefore, to be clarified, the sentence was now modified as follows:

$H^*_\mathrm{HNO_3}$ is the effective Henry's law constant (products of Henry's law constant for $\mathrm{HNO_3}$ gases and the acid dissociation constant for $\mathrm{HNO_3} \leftrightarrow \mathrm{NO_3^-} + \mathrm{H^+}$) dependent on temperature and pH ($\mathrm{mol^2\ kg^{-2}\ atm^{-1}}$ on a molality-basis; Clegg et al., 1998).

l. 392: the increase from 3 to 22 $\mu$g m$^{-3}$ is closer to 7 times than 8 times. Please replace '8' by '7' accordingly.

It was now corrected.

Fig. S5: What are the blue and red frames?

The figure caption was modified as follows:

**Figure S6: Estimated annual NH$_3$ emissions in Seoul for 2008–2015 from the Clean Air Policy Support System (CAPSS) inventory (NIER, 2018). Blue and red frames represent the years of measurement in Phan et al. (2013) and this study, respectively.**

[revised manuscript text omitted]